



# Speleothem Evidence for Megadroughts in the SW Indian Ocean during the Late Holocene

Hanying Li[1], Hai Cheng[1,2*], Ashish Sinha[3], Gayatri Kathayat[1], Christoph Spötl[4], Aurèle Anquetil André[5], Arnaud Meunier[5], Jayant Biswas[6], Pengzhen Duan[1], Youfeng Ning[1], R. Lawrence Edwards[2]

[1] Institute of Global Environmental Change, Xi'an Jiaotong University, China
[2] Department of Earth Sciences, University of Minnesota, Minneapolis, USA
[3] Department of Earth Science, California State University Dominguez Hills, Carson, USA
[4] Department of Earth and Environmental Sciences, University of Illinois, Chicago, USA
[5] Francois Leguat Giant Tortoise and Cave Reserve, Anse Quitor, Rodrigues Island, Mauritius.
[6] National Cave Research and Protection Organization, Raipur, 492001, India.

*Correspondence and requests for materials should be addressed to H.C. (email: cheng021@xjtu.edu.cn)

## Abstract

The '4.2 ka BP event' is widely described as a 200-300 years long interval of major climate anomaly (typically, arid and cooler conditions potentially across the globe), which defines the beginning of the current 'Meghalayan' age in the Holocene epoch. The 4.2 ka event however, has been
disproportionately reported from proxy records situated at low-mid latitudes in the Northern Hemisphere. Consequently, the climatic manifestation of the 4.2 ka event in both spatial and temporal domains is still much less clear in Southern Hemisphere. This is particularly the case for the southwest sector of the southern Indian Ocean. Here we present high-resolution and chronologically well-constrained speleothem oxygen and carbon isotopes records of hydroclimate variability between ~6 and
3 ka ago from Rodrigues Island, located in the southwest subtropical Indian Ocean, ~ 600 km east of Mauritius. Our records reveal a major shift to drier condition at circa 4 ka BP, which culminated into a multicentennial period of drought (i.e., megadrought) that lasted continuously from ~ 3.9 to 3.5 ka BP. The inferred hydroclimatic conditions between 4.0 and 4.2 ka BP, are however not distinctly distinguishable from the region's mean hydroclimatic state over the length of our record. Because the
precipitation variability at Rodrigues is distinctly modulated by meridional movement of the Inter-Tropical Convergence Zone and the El Niño Southern Oscillation dynamics, our proxy data may ultimately provide critical constraints in our understanding the timing and dynamical forcing of the 4.2 ka event.






## 1. Introduction

The '4.2 ka event' is a widespread climate event that occurred between ~ 4.2 and 3.9 ka BP (thousand years before present, where present =1950 AD) (e.g., Weiss et al., 2016). Many paleoclimate records from Northern Hemisphere (NH) have characterized the event as a multidecadal-multicentennial period of arid and cooler conditions across the Mediterranean, the Middle East, South Asia and North Africa (e.g., Finné et al., 2011; Marchant and Hooghiemstra, 2004; Migowski et al., 2006; Mayewski et al., 2004; Staubwasser et al., 2003; Arz et al., 2006; Zielhofer et al., 2017; Stanley et al., 2003; Kathayat et al., 2017). While some Southern Hemisphere (SH) proxy records from the tropical and sub-tropical regions of Africa and South America also show a shift to a drier climatic regime around 4 ka BP (e.g., Thompson et al., 2002; Russell et al., 2003; Marchant & Hooghiemstra, 2004), many other proxy records also show spatially complex and heterogeneous climate patterns around this time (e.g., Mayewski et al., 2004; Wanner et al., 2008; Russell et al., 2003; Berke et al., 2012; Tierney et al., 2011; Verschuren et al., 2009; de Boer et al., 2013, 2014, 2015; Rijsdijk et al., 2009, 2011; Tierney et al., 2008).

The driving mechanisms of the 4.2 ka event also remain elusive. For example, there is no clear evidence of rapid freshwater injection in the north Atlantic that could have disrupted the thermohaline circulation, and thereby, produced attendant changes in global climate in a manner akin to the 8.2 ka BP event (e.g., Walker et al., 2012). There are also no evidences of major perturbations in atmospheric concentrations of aerosols and $CO_2$ around this time (Monnin et al., 2001). A southward meridional shift in the mean position of the Inter-Tropical Convergence Zone (ITCZ) during this time has also been hypothesized (Mayewski et al., 2004), which potentially can account for the low–latitude aridity in many NH locations but this hypothesis is inconsistent with results from some proxy records situated at the southern margin of the ITCZ in SH, which show little or no evidence of wetter condition during this time (e.g., Russell et al., 2003; de Boer et al., 2013, 2014, 2015; Rijsdijk et al., 2009, 2011; Berke et al., 2012). Another hypothesis calls for a switch-on of the modern El Niño Southern Oscillation (ENSO) regime (e.g., Donders et al., 2008; Conroy et al., 2008), which became more pronounced in the mid-latitude regions after ca. 4 ka BP (e.g., Barron and Anderson, 2010). Although a marked asymmetrical increase in the ENSO variance (i.e., more frequent El Niño events) can account for the weakened Asian monsoon during the 4.2 ka BP event, but what impacts it may have had on the mid-latitudes climates in both hemisphere is not clear.

Here we present two precisely dated speleothem oxygen ($\delta^{18}O$) and carbon ($\delta^{13}C$) isotope records from La Vierge cave (LAVI-4) and Patate cave (PATA-1) from Rodrigues Island (Fig.1), located in the southern subtropical Indian Ocean. The LAVI-4 and PATA-1 records span from 6.0 to 3.0 ka BP and 6.1 to 3.5 ka BP, with average resolutions of ~4 and ~14 years, respectively. Our new records have tight



age control and sub-decadal resolution, which allow us to characterize hydroclimate variations in the southwestern Indian Ocean between 3 and 6 ka BP interval.

## 2 Cave Location and Modern Climatology

### 2.1 Climatology

Rodrigues (~19°42'S, ~63°24'E) is a small volcanic island (~120 km$^2$) situated in the southwestern Indian Ocean, ~600 km east of Mauritius (Fig. 1). The island's maximum altitude is ~400 m above sea level. Rodrigues' mean annual temperature is ~24°C and the mean annual rainfall is ~1010 mm, of which nearly 70% occurs during the wet season (November to April) with February being the wettest month. The seasonal distribution of rainfall is largely controlled by the seasonal migration of the

ITCZ and the Mascarene High (Senapathi et al., 2010; Rijsdijk et al., 2011; Morioka et al., 2015) (Fig. 1). Given its location at the southern fringe of the ITCZ, the austral summer rainfall at Rodrigues is sensitive to the mean position of the southern limit of the ITCZ. This is highlighted by backward (~120 hours) HYSPLIT (Draxler and Hess, 1998) trajectory composites of the low level winds (850 hPa) for years when the total January to March (JFM) precipitation was unusally low (dry) and high (wet) than

the long-term mean (1951-2016) at Rodrigues (Fig. 1B). Of note is a major increase in fraction of air parcel trajectories arriving from the north of Rodrigues during the wetter years, indicating an enhanced contribution of northerly moisture resulting from a more southern position of the ITCZ (Fig. 1B). This observation is further supported by low level wind trajectory clusters composites for the month of February for those years when the southern boundary of the the ITCZ was anomolusly north or south

(Lashkari et al., 2017; Freitas et al., 2017) of its long-term mean February position (Fig. 2 A-B). In addition to the ITCZ, the ENSO dynamics also modulate austral summer precipiation in Rodrigues via modulating the Hadley and Walker circulations. (Senapathi et al., 2010; de Boer et al., 2014; Griffiths et al., 2016; Zinke et al., 2016). Instrumental data and our trajectory composites for select El Niño and La Niña years suggest that an increased (decreased) summer precipitation at Rodrigues are associated with

the El Niño (La Niña) events (Fig. 2 C-D).

### 2.2 Oxygen isotope climatology

    The modern observations of $\delta^{18}O$ of precipitation ($\delta^{18}O_p$) in the study area are unavailable due to lack of Global Network of Isotopes in Precipitation (GNIP) stations in Rodrigues. However, $\delta^{18}O_p$ data from the nearest GNIP station in Mauritius show a clear annual cycle in $\delta^{18}O_p$ with depleted values

during the austral summer (Fig. 3A). Additionally, in absence of GNIP data, we use simulated $\delta^{18}O_p$ data from the Experimental Climate Prediction Center's Isotope-incorporated Global Spectral Model (IsoGSM) (Yoshimura et al., 2008) to assess the large-scale dynamical processes that control $\delta^{18}O_p$ on interannual and decadal timescales. Model results show the presence of a strong negative correlation between the $\delta^{18}O_p$ and rainfall amount similar to the 'amount effect' (e.g., Dansgaard, 1964) (Fig. 3 B-



C). We therefore, interpret $\delta^{18}O_p$ variations in the cave catchment and consequently, in speleothems from this region to primarily reflect variations in the rainfall amount in response to both local and large-scale atmospheric circulation changes in such that more negative (positive) $\delta^{18}O_p$ values occur during times of anomalous southward (northward) position of the southern boundary of the ITCZ and the El Niño (La Niña) conditions, respectively.

## 3 Methods and Results

### 3.1 Speleothem Samples

    Two stalagmite samples, LAVI-4 and PATA-1, from La Vierge cave and Patate cave respectively, are used in this study. La Vierge (19°45′26″S, 63°22′13″E, ~32 masl) and Patate (19°45′30″S, 63°23′11″E, ~20 masl) caves are located in Plaine Corail and Plaine Caverne in southwestern Rodrigues, respectively (Middleton and Burney, 2013). The cave temperature and relative humidity at the time of sample collection (June 2015) were ~25.5°C and 95% in La Vierge cave and ~22.5°C and 95% in Patate cave, respectively. Samples LAVI-4 and PATA-1 were collected at the distance of ~50 m and 200 m from cave entrances, respectively. The diameters of LAVI-4 and PATA-1 are ~75 and 95 mm, and the length ~400 and ~334 mm, respectively. Both stalagmite samples were cut along their growth axes, using a thin diamond blade, and then polished. Both samples grew continuously between 3.5 and 6.0 ka BP interval without any visible hiatuses (Fig. 4).

### 3.2 $^{230}$Th Dating

    Subsamples (~80-130 mg) for $^{230}$Th dating from LAVI-4 and PATA-1 were drilled using a 0.9 mm carbide dental drill. The $^{230}$Th dating was performed at Xi'an Jiaotong University, China by using Thermo-Finnigan Neptune-*plus* multi-collector inductively coupled plasma mass spectrometers (MC-ICP-MS). The methods were described in Cheng et al. (2000, 2013). We used standard chemistry procedures (Edwards et al., 1987) to separate U and Th. A triple-spike ($^{229}$Th–$^{233}$U–$^{236}$U) isotope dilution method was used to correct instrumental fractionation and to determine U/Th isotopic ratios and concentrations (Cheng et al., 2000, 2013). U and Th isotopes were measured on a MasCom multiplier behind the retarding potential quadrupole in the peak-jumping mode using the standard procedures (Cheng et al., 2000). Uncertainties in U and Th isotopic measurements were calculated offline at 2σ level, including corrections for blanks, multiplier dark noise, abundance sensitivity, and contents of the same nuclides in spike solution. $^{234}$U and $^{230}$Th decay constants are reported in Cheng et al. (2013). Corrected $^{230}$Th ages assume the initial $^{230}$Th/$^{232}$Th atomic ratio of 4.4 ±2.2 ×10$^{-6}$, the values for material at secular equilibrium with the bulk earth $^{232}$Th/$^{238}$U value of 3.8. The corrections for a few dating results of sample LAVI-4 and PATA-1 are large because either the U concentrations are relatively low (~65 ppb) or the detrital $^{232}$Th concentrations are relatively high (>100 ppt) (Table S1, Fig. 5).



### 3.3 Age Models

We obtained 26 and 5 $^{230}$Th dates for samples LAVI-4 and PATA-1, respectively. The LAVI-4 and PATA-1 age models and associated uncertainties were constructed using COPRA (Constructing Proxy Records from Age) (Breitenbach et al., 2012) and ISCAM (Fohlmeister, 2012) age modelling schemes (Fig. 4). Both modelling schemes yielded virtually identical age models, and thus conclusions of this study are not sensitive to the choice of different age models (Figs. 4 and 5).

### 3.4 Stable Isotope Analysis

LAVI-4 and PATA-1 stable isotope ($\delta^{18}$O and $\delta^{13}$C) records are established by ~952 and ~192 data, respectively. The New Wave Micromill, a digitally controlled tri-axial micromill equipment, was used to obtain subsamples. The subsamples (~80 µg) were continuously micromilled from LAVI-4 and PATA-1 with typical increments between of 200 and 100 µm along the stalagmites growth axes,
respectively. The subsamples of LAVI-4 were measured using Finnigan MAT-253 mass spectrometer coupled with an on-line carbonate preparation system (Kiel-IV) in the Isotope Laboratory, Xi'an Jiaotong University. The subsample of PATA-1 were measured using an on-line carbonate preparation system (GasbenchII) connected to an isotope ratio mass spectrometer (Delta$^{plus}$XL) in the Isotope Laboratory, Innsbruck University. The technique used in the Innsbruck University are reported in Spötl
(2011) and Spötl and Vennemann (2003). All results are reported in per mil (‰) relative to the Vienna PeeDee Belemnite (VPDB) standard. Duplicate measurements of standards show a long-term reproducibility of ~0.1‰ (1σ) or better (Table S2, Fig. 5).

### 3.5 Isotopic Equilibrium Test

Conventional criteria for stalagmite isotopic equilibrium assessment is the Hendy Test (Hendy,
1971), which require weak or no correlation between speleothem $\delta^{18}$O and $\delta^{13}$C values measured along the growth axis as well as along the same growth lamina. The correlation between the $\delta^{18}$O and $\delta^{13}$C values in LAVI-4 and PATA-1 are 0.53 and 0.85, respectively, which on the basis of the Hendy Test, may indicate disequilibrium. However, some studies (e.g. Dorale and Liu, 2009) suggest that weak correlation between $\delta^{18}$O and $\delta^{13}$C values is not a prerequisite to isotopic equilibrium. Instead, the
replication test (i.e., a high degree of coherence between individual $\delta^{18}$O profiles of different speleothems from the same cave over the period of overlap) is a more stringent and reliable test of isotopic equilibrium. Indeed, a high degree of visual similarity between the coeval portions of LAVI-4 and PATA-1 $\delta^{18}$O and $\delta^{13}$C records suggest that both the stalagmites record primary climate signal, notwithstanding the offsets between the absolute values in two records (Fig. 5A). This is also confirmed
by statistically significant correlations between the LAVI-4 and PATA-1 $\delta^{18}$O (r =0.61 at 95% confidence level) and $\delta^{13}$C (r =0.69 at 95% confidence level) records obtained through the ISCAM (Intra-Site Correlation Age Modelling) algorithm (Fohlmeister, 2012) during their contemporary growth





period between ~3.4–6.0 ka BP (Fig. 5). ISCAM strives to find the best correlation between the proxy records by adjusting each record within the margin of dating uncertainty by using a Monte Carlo approach. The significant levels are assessed against a red-noise background from generating artificially simulated first-order autoregressive time series (AR1). The offset in absolute $\delta^{18}O$ values between the LAVI-4 and PATA1 however, remains unclear, possibly arising from processes related to different karstic characteristics of the two caves. Therefore, in the following discussion we focus only the LAVI-4 $\delta^{18}O$ and $\delta^{13}C$ records due to their higher resolution, robust chronology and high-fidelity.

## 4 Discussion and Conclusions

### 4.1 Proxy Interpretations

The temporal resolution of LAVI-4 $\delta^{18}O$ profile between 3 and 6 ka BP varies from 1.2 to 16.4 years with an average resolution of ~3.2 years and is characterized by a large (~4.0 ‰) variability. As noted earlier, we interpret temporal variations in the LAVI-4 profile to dominantly reflect changes in the precipitation amount. This line of reasoning is justified given the island's isolated setting far removed from large-sized landmasses and its low topographic relief, which minimizes isotopic variability stemming from processes such as the continentality and altitude effects as well as mixing of distant water vapor sources with significantly different isotopic compositions. This interpretation is additionally supported by moderate to strong covariance between the LAVI-4 $\delta^{18}O$ and $\delta^{13}C$ profiles. Although the temporal variations in the latter can stem from changes in vegetation type and density, soil microbial productivity, prior calcite precipitation (PCP) and ground water infiltration rates (e.g., Baker et al., 1997; Genty et al., 2003), all of which may drive $\delta^{18}O$ and $\delta^{13}C$ values in the same fashion (e.g., Brook et al., 1990; Dorale et al., 1992; Bar-Matthews and Ayalon, 1997). The significant covariance between $\delta^{13}C$ and $\delta^{18}O$ records could therefore, indicate that both proxies reflect a common response to changes in rainfall amount at Rodrigues.

### 4.2 Hydroclimate Variability between 6 and 3 ka BP at Rodrigues

The LAVI-4 $\delta^{18}O$ and $\delta^{13}C$ records from 6 to 3 ka BP show prominent multidecadal to multicentennial scale variability but no clear long-term trends (Fig. 5 and 6). The z-score transformed profile of LAVI-4 $\delta^{18}O$ profile reveals several decadal to multidecadal intervals of significantly drier and wetter conditions (> ±1 standard deviation) (Fig. 6) that occurred throughout the length of the record. One of the most prominent features of our record is a switch from an interval characterized by $\delta^{18}O$ high-frequent variance (i.e., 6 to 4 ka BP) to the one with progressively higher $\delta^{18}O$ values from ~4 to 3.55 ka BP, suggesting a prolonged multi-centennial megadrought occurred in Rodrigues. After 3.54 ka BP, the $\delta^{18}O$ sharply decreased again until 3.12 ka BP, indicating the abrupt end of the megadrought (Fig. 6). The LAVI-4 $\delta^{13}C$ record shows a variation pattern broadly similar with the $\delta^{18}O$ record and delineates three major drought events between 6 and 3 ka, centered at 5.43, 4.62 and 3.54 ka



BP respectively, ~ 800-1000 years apart. All the three drought events show a saw-tooth pattern characterized by a long-term gradual positive excursion followed by an abrupt termination (Fig. 6). The interval corresponding to the '4.2 ka event', typically considered between 4.2 and 3.9 ka BP (e.g., Weiss
et al., 2016), in the LAVI-4 records does not however, stand out as 'pulse-like' event as evident in many other proxy records. Instead, as noted above, it marks the onset of a multicentennial trend towards drier condition near the end of the typical '4.2 ka event' interval.

**4.3 Broader Spatial Patterns of Hydroclimate Variability between 6 and 3 ka BP**

The most prominent interval of drier hydroclimatic condition in our record between 3.9 and 3.5 is
also evident in other records from Southeast Africa, such as records from Lake Edward (Russell et al., 2003), Lake Victoria (Berke et al., 2012), Zambezi delta (Schefuß et al., 2011) and Tatos basin (de Boer et al., 2014) (Fig. 7). Additionally, the record from Lake Malawi, East Africa (Konecky et al., 2011) also shows a weak dry excursion between 4.1 to 3.5 ka BP in the context of continuously wet trend between 6 and 3 ka BP. In the eastern sector of the southern Indian Ocean, speleothem records from
Sumatra, Indonesia (Wurtzel et al., 2018), Northwest Australia (Denniston et al., 2013), and Liang Luar cave (Griffiths et al., 2009) also show the onset of generally drier conditions from ~ 4 ka BP (Fig. 8).

Previously, multiple hypotheses have been proposed to explain the onset of drier conditions ~ 4 ka BP in the southwestern Indian Ocean and Southeast Africa, which include shifts in the mean position of the ITCZ (Russell et al., 2005; Railsback et al., 2018), changes in the sea surface temperature (SST)
gradient between western and eastern Indian Ocean (Indian Ocean Dipole, IOD) (Berke et al., 2012; de Boer et al., 2014), and increase in the ENSO variance (Tierney et al., 2011; de Boer et al., 2014). If IOD or ENSO is the main driving mechanism, an anti-phased relation between the Rodrigues record in the west and proxy record from eastern margin of the southern Indian Ocean including northern Australia would be expected (e.g., see the spatial pattern of El Niño related precipitation anomalies in Fig. 2D),
which is inconsistent with phase relationship presented in the Fig. 8. As such therefore, IOD and ENSO does not readily explain the observed relationships between Rodrigues and other sites across the southern Indian Ocean.

A progressive trend towards drier condition in Rodrigues between ~4 and 3.5 ka BP suggest a northward shift in the mean position of the ITCZ (e.g., Fig. 2A). This inference, if correct, stands in
direct contrast with the postulated southward shift in the ITCZ, often invoked to explain the weakening of the Asian monsoons since ~ 4.2 ka BP (e.g., Kathayat et al., 2017; Wang et al., 2005). Consequently, drier conditions on both northern and southern margins of the ITCZ in both hemispheres, argue against the idea of a simple southward shift in the mean position of the ITCZ as a probable cause of the 4.2 ka event. A more probable mechanism may therefore involve an overall contraction in the north-south
range of the migrating ITCZ belt (e.g., Yan et al., 2015; Denniston et al., 2016; Scroxton et al., 2017) together with its reduced width which appears to better explain the spatial pattern of hydroclimate





change observed in both hemispheres. For example, between ~4 and 3.5 ka BP, basin wide drier conditions occurred near the contemporary southern limit of the austral summer ITCZ in the southern Indian Ocean as indicated by proxy records from Zambezi Delta (Schefuβ et al., 2011), Tatos Basin (de

Boer et al., 2014), Rodrigues and northwest Australia (Denniston et al., 2013) (Figs. 7 and 8). Additionally, a high resolution record from Sahiya cave (Kathayat et al., 2017), located at the northern limit of the boreal summer ITCZ, also show a synchronously drying trend between 4.0 and 3.5 ka BP (Fig. 9).

Additionally, in parallel with drier condition in the southern limit of the austral summer ITCZ,

proxy records from Lake Edward (Russell et al., 2003), Lake Victoria (Berke et al., 2012) and Tangga cave (Wurtzel et al., 2018), which lie near the north limit of the contemporary austral summer ITCZ also exhibit dry conditions. In contrast to the dry condition on both northern and southern limits of the austral summer ITCZ, records within the core location of the austral summer ITCZ, such as Lake Challa (Tierney et al., 2011), Lake Tanganyika (Tierney et al., 2008), Lake Malawi (Konecky et al., 2011) and

Makassar Strait (Tierney et al., 2012), show either slightly wetter or virtually unchanged hydroclimatic condition (Figs. 7 and 8). Based on these observations, we suggest that the ITCZ contraction in terms of both north-south meridional shift as well as in its overall width may have played an important role in producing the hydroclimatic changes in our study area. Furthermore, changes in the strength of the SH westerly winds (SHW) (Marx et al., 2011; Saunders et al., 2018) also appear to correlate to the

megadrought event observed in the Rodrigues record (Fig. 7), which may have restricted the southern range of the austral summer ITCZ.

## 5 Author Contributions

H.C., A.S. and H.Y.L designed the research and experiments; H.C., A.S., J.B., Y.F.N. and H.Y.L. completed the fieldwork; H.Y.L., H.C., Y.F.N. and C.S. performed all stable isotope measurements and

[230]Th dating work. A.S., L.Y. and H.Y.L. did the data analyses. H.C., H.Y.L. and A.S. wrote the manuscript, with the help of all other co-authors.

## 6 Competing interests

The authors declare no competing financial interests.

## 7 Acknowledgments

This work was supported by grants from NSFC (41472140, 41731174 and 41561144003); US NSF grant 1702816; and a grant from State Key Laboratory of Loess and Quaternary Geology, Institute of Earth Environment, CAS (SKLLQG1414).





## 8 Data and materials availability

All data needed to evaluate the conclusions in the paper are presented in the paper. Additional data related to this paper may be requested from the authors. The data will be archived at the National Climate Data Center (https://www.ncdc.noaa.gov/data-access/paleoclimatology-data). Correspondence and requests for materials should be addressed to H.C. (cheng021@xjtu.edu.cn).

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





**Figures:**

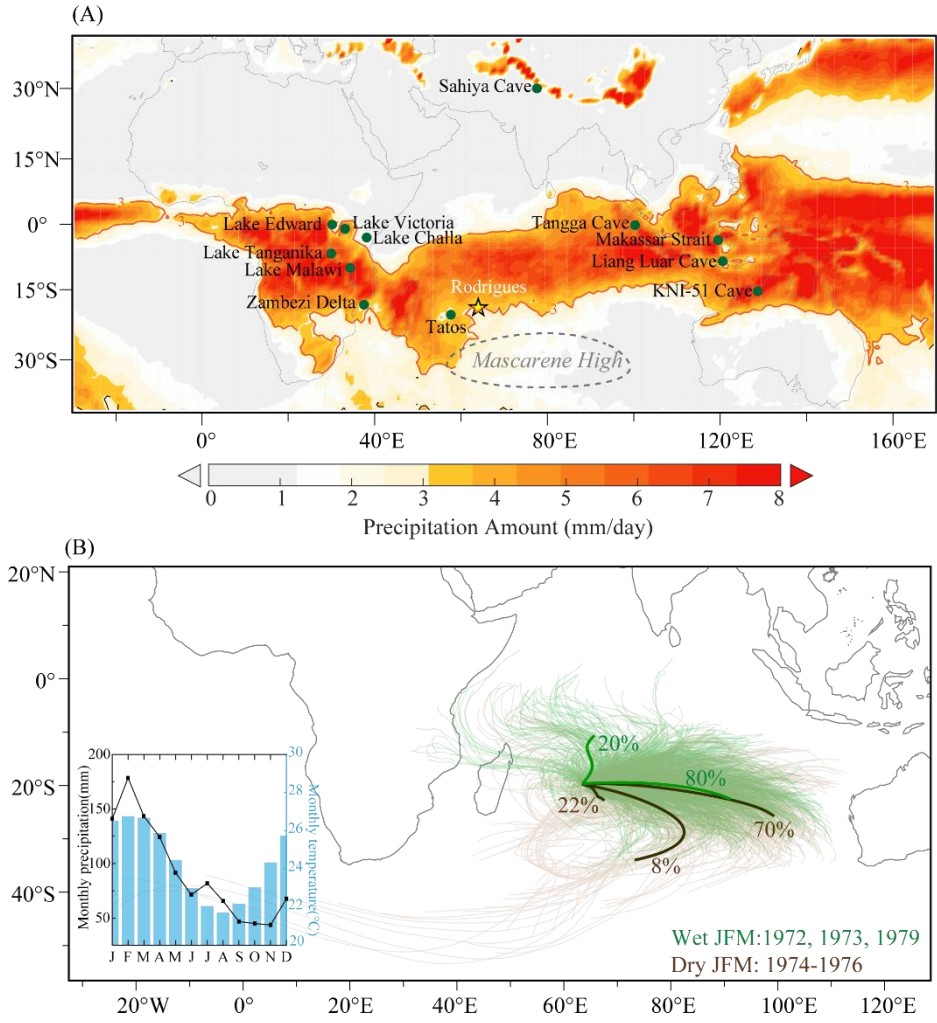

**Figure 1. Proxy locations and climatology. (A)** The mean January to March (JFM) precipitation from
the Tropical Rainfall Measuring Mission (TRMM) (https://trmm.gsfc.nasa.gov/) averaged over the
period from 1997 to 2014. Shaded area bounded by solid red lines (3 mm day$^{-1}$ isohyet) depict the mean
position of the ITCZ. The dashed line shows the mean position of JFM 850 hPa geopotential height
marking the location of the Mascarene High. Locations of Rodrigues Island (yellow star, this study) and
other proxy sites (green dots) discussed in the text are also shown. **(B)** 4x daily low-level (~850 hpa)
JFM air parcel back (120 hours) trajectory composites for anomalously wet (green) and dry (brown)
years. Trajectories were computed using NOAA HYSPLIT model (Draxler and Hess, 1998) using
NCEP/NCAR Reanalysis data (Kalnay et al., 1996). Bold lines indicate main cluster tracks associated
with trajectories for wetter (green) and drier (brown) years. Inset shows mean monthly rainfall and
temperature at Rodrigues averaged over the period from 1951 to 2015.





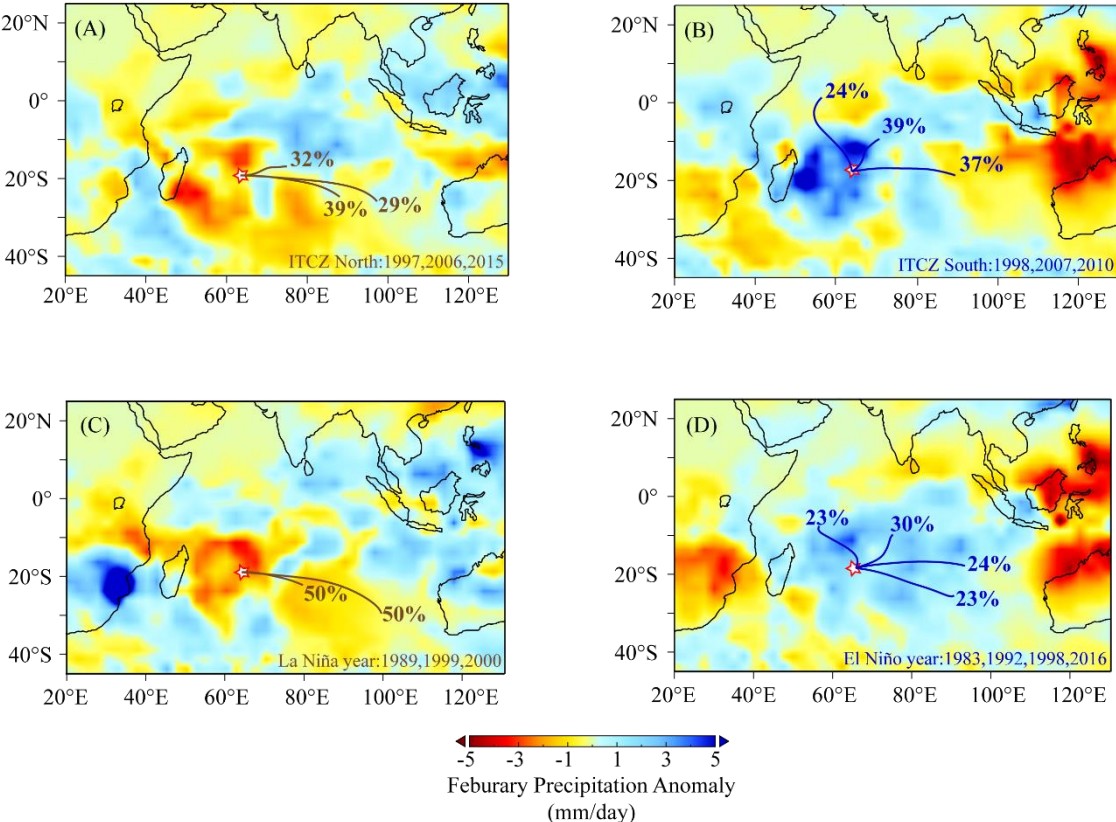


**Figure 2. The ITCZ and ENSO dynamics. (A and B)** The spatial composite maps of precipitation anomalies for February (anomalies calculated with respect to the period 1981-2010) for the years marked by anomalous northward (A, 1997, 2006, 2015) and southward (B, 1998, 2007, 2010) location of the southern boundary of the ITCZ (Lashkari et al., 2017; Freitas et al., 2017). The maps are overlaid
by backward (120 hours) low level air parcel trajectory clusters and their relative contributions. **(C and D)** Same as in **A** and **B** but for the La Niña (C, 1989, 199, 2000) and El Niño (D, 1983, 1992, 1998, 2016) years. Precipitation data is from GPCP (Adler et al., 2018).





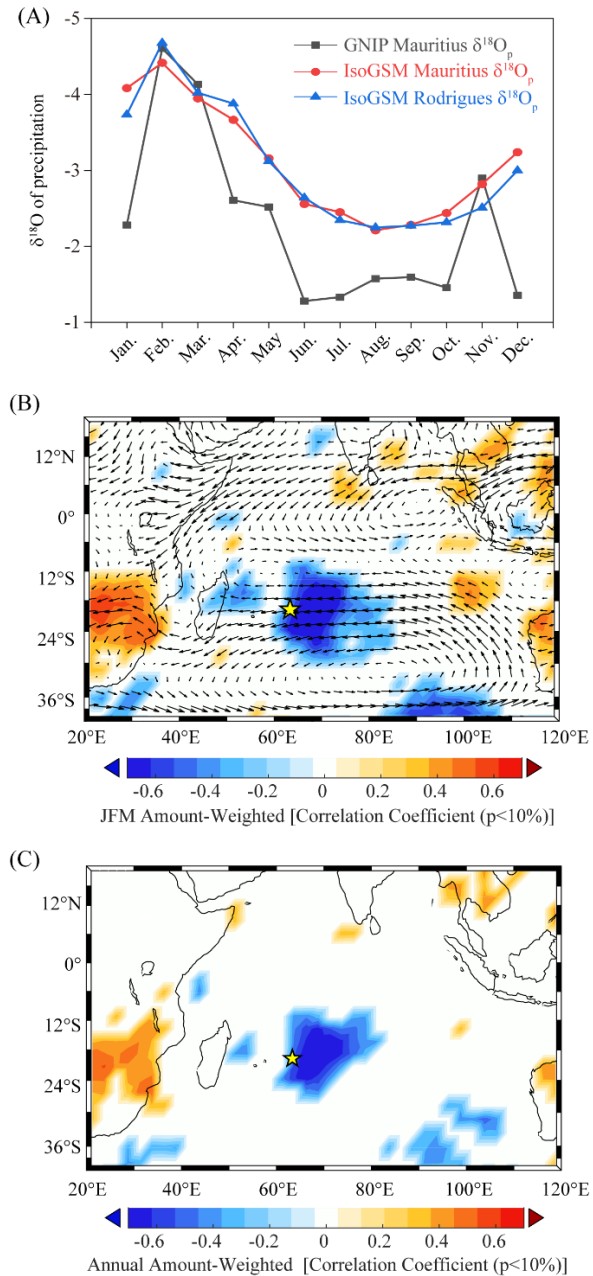

**Figure 3. Model and observational data of δ¹⁸O in precipitation in the study area**. (**A**) Monthly means of simulated $\delta^{18}O_p$ from Mauritius (red) and Rodrigues (blue) from IsoGSM (Yoshimura et al., 2008). Also shown is monthly means of $\delta^{18}O_p$ from six GNIP stations in Mauritius (black) covering the periods from 1992-1995 and 2009-2014. (**B and C**) The spatial correlation maps for JFM (**B**) and annual (**C**) amount-weighted IsoGSM $\delta^{18}O_p$ from the nearest grid point to Rodrigues and the GPCP precipitation (GPCP v2.3) (Adler et al., 2018) for the period 1979 to 2016.




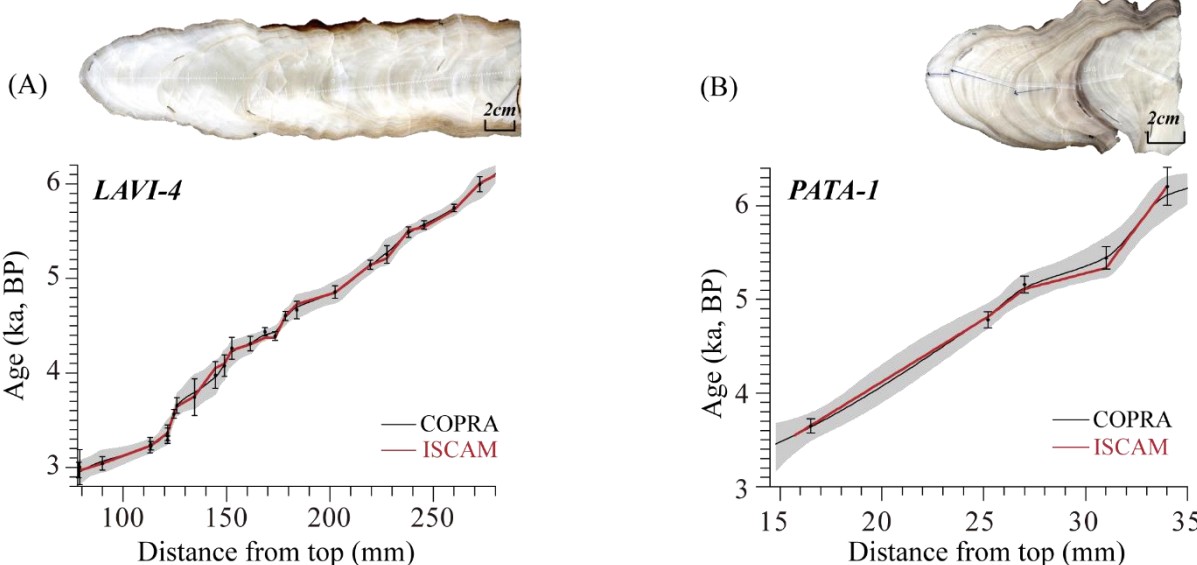

**Figure 4. Age models of LAVI-4 and PATA-1 records. (A)** LAVI-4 age models and age uncertainties using COPRA (Breitenbach et al., 2012) (black) and ISCAM (Fohlmeister, 2012) (red) modeling methods. The gray band depicts the 95% confidence interval from COPRA. Error bars on $^{230}$Th dates represent 2σ analytical errors. **(B)** Same as in **(A)** but for sample PATA-1.





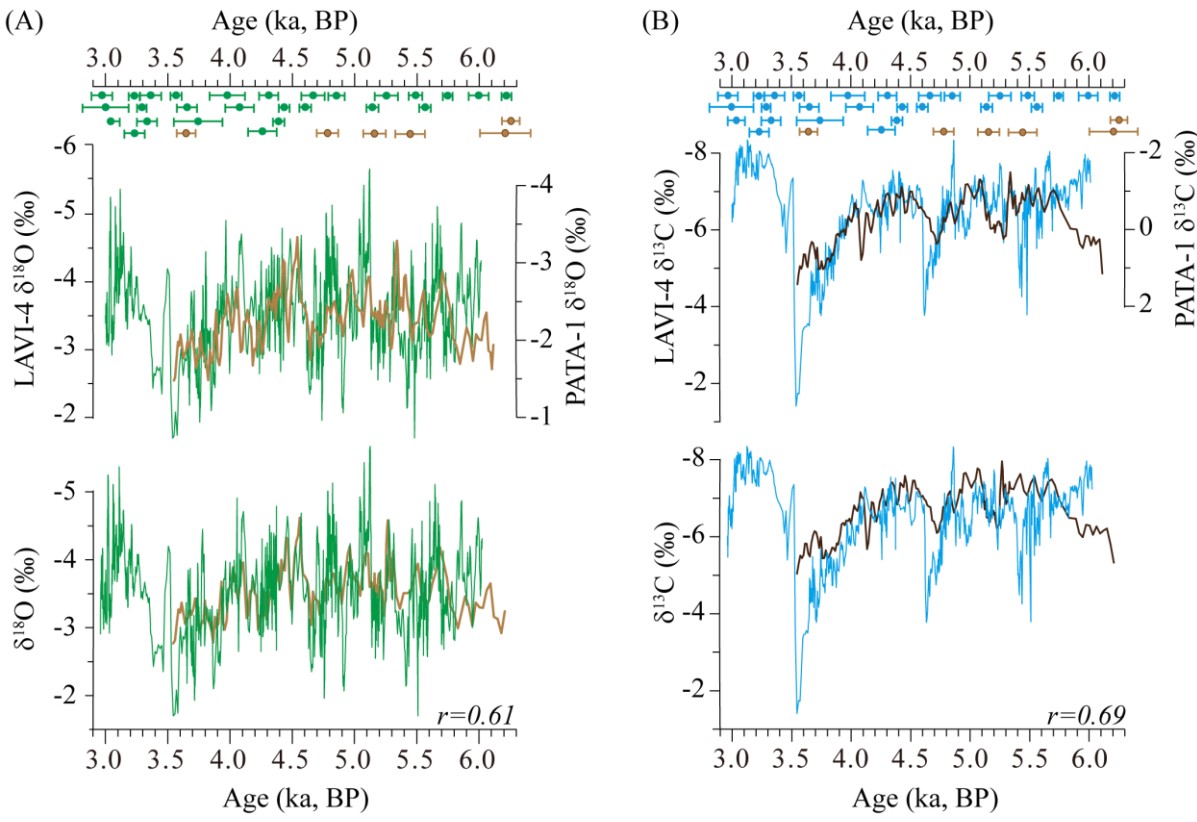

**Figure 5. δ¹⁸O and δ¹³C records of LAVI-4 and PATA-1.** **(A)** The δ¹⁸O profiles of LAVI-4 (green) and PATA-1 (brown) on their independent COPRA (Breitenbach et al., 2012) age models (top) and ISCAM (Fohlmeister, 2012) derived age models (bottom). The correlation coefficient (*r*) between LAVI-4 and PATA-1 is *0.61*. The PATA 1 δ¹⁸O values were adjusted by ~1.5 ‰ to match with the LAVI-4. **(B)** Same as in **(A)** but for the δ¹³C profiles of LAVI-4 and PATA-1. The PATA-1 δ¹³C values were adjusted by ~6.7 ‰ to match with the LAVI-4 values.





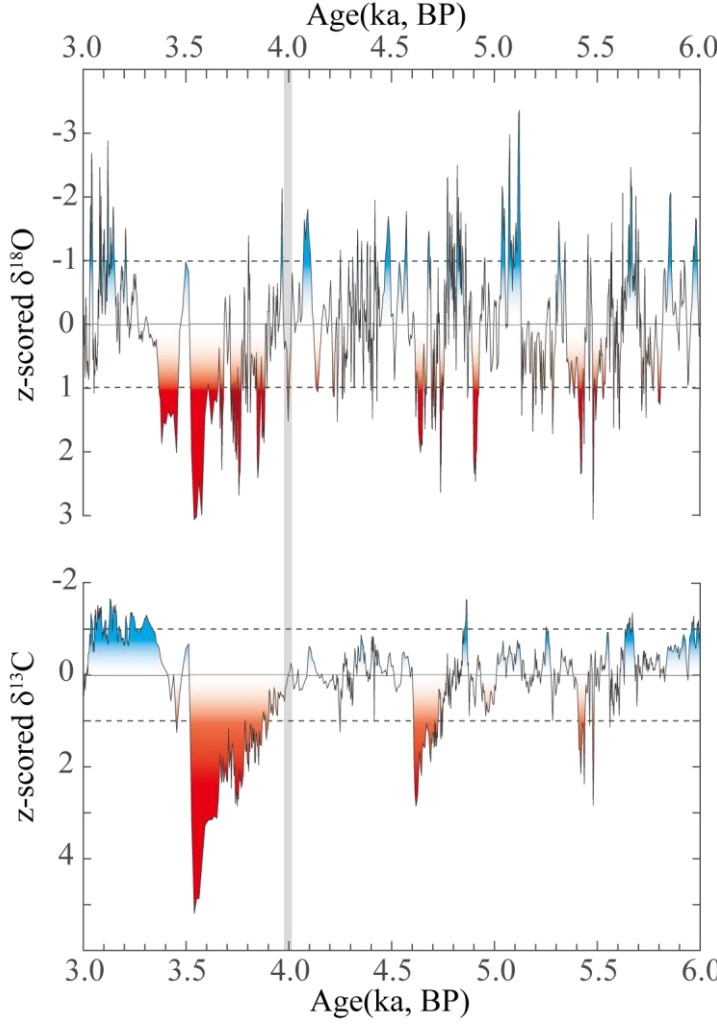

**Figure 6. The inferred hydroclimatic variability at Rodrigues from 6 to 3 ka BP.** The LAVI-4 $\delta^{18}O$ and $\delta^{13}C$ record shown as z-score transformed. Inferred drought (z-score > 1) and pluvial (z-score < -1) are shaded (increasing saturation index indicates increasing intensity). Dash lines indicate the 1 standard deviation. Grey bar at ~4 ka BP mark the beginning of multi-centennial period of drier condition.



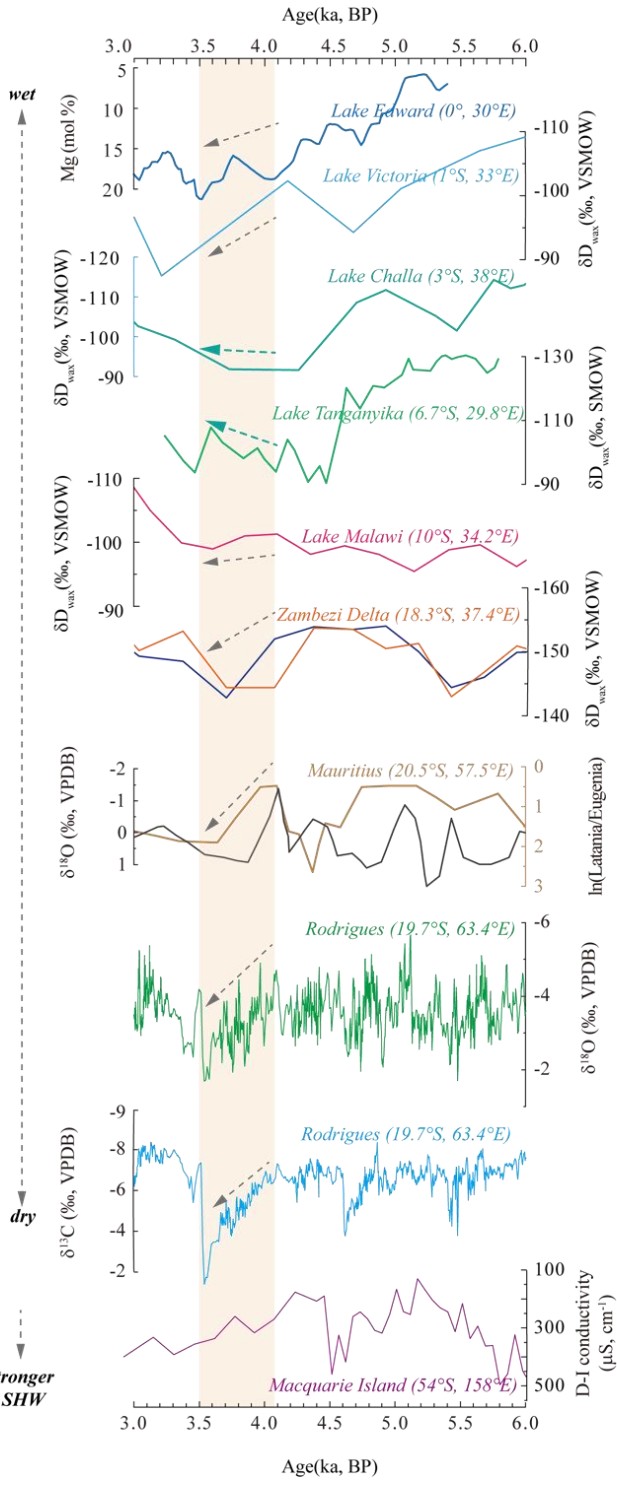

**Figure 7. Comparison of LAVI-4 with climate proxy records from East Africa and Macquarie Island.** From top to bottom, Mg concentration of sedimentary endogenic calcite from Lake Edward

...





(Russell et al., 2003); the δD$_{leaf\ wax}$ records from Lake Victoria (Berke et al., 2012); Lake Challa
535 (Tierney et al., 2011); Lake Tanganyika (Tierney et al., 2008); Lake Malawi (Konecky et al., 2011); the
δD of the n-C$_{29}$ alkane (dark blue) and the n-C$_{31}$ alkane (orange) from Zambezi delta (Schefuβ et al.,
2011); the δ$^{18}$O record (black) and ln (Latania/Eugenia) records (brown) from Tatos basin, Mauritius
(de Boer et al., 2014); the LAVI-4 δ$^{18}$O and δ$^{13}$C record from La Vierge cave (this study); and the D-I
conductivity from Lake Emerald, Macquarie Island (Saunders et al., 2018). Shaded vertical bar marks
540 the duration from ~ 4.1 to 3.5 ka BP. Grey and green dashed arrows mark the drying and wet trend
inferred from East Africa lake records, respectively. All y axes (except for the Macquarie island profile)
are inverted to show drier conditions down.





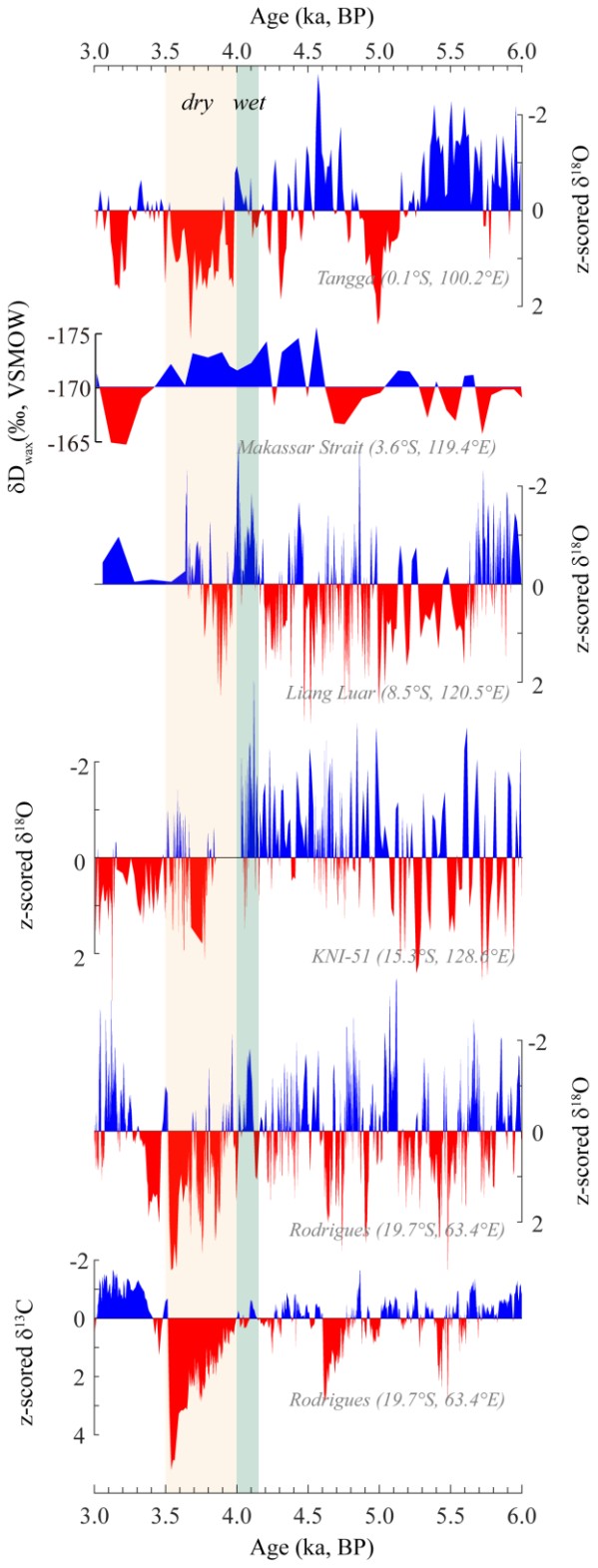





**Figure 8. Comparison of LAVI 4 with climate proxy records from the eastern Indian Ocean.** From top to bottom, z-score transformed speleothem $\delta^{18}O$ record from Tangga cave, Sumatra, Indonesia (Wurtzel et al., 2018); the $\delta D_{leaf\ wax}$ record from marine sediment core BJ8-03-70GGC in the Makassar Strait (Tierney et al., 2012); z-score transformed speleothem $\delta^{18}O$ records from Ling Luar western Flores, Indonesia (Griffiths et al., 2009); KNI-51 cave, Kimberley, northwestern Australia (Denniston et al., 2013); and LAVI-4 $\delta^{18}O$ and $\delta^{13}C$ records from La Vierge cave (this study). Shaded vertical bars mark periods of drier and wetter conditions.

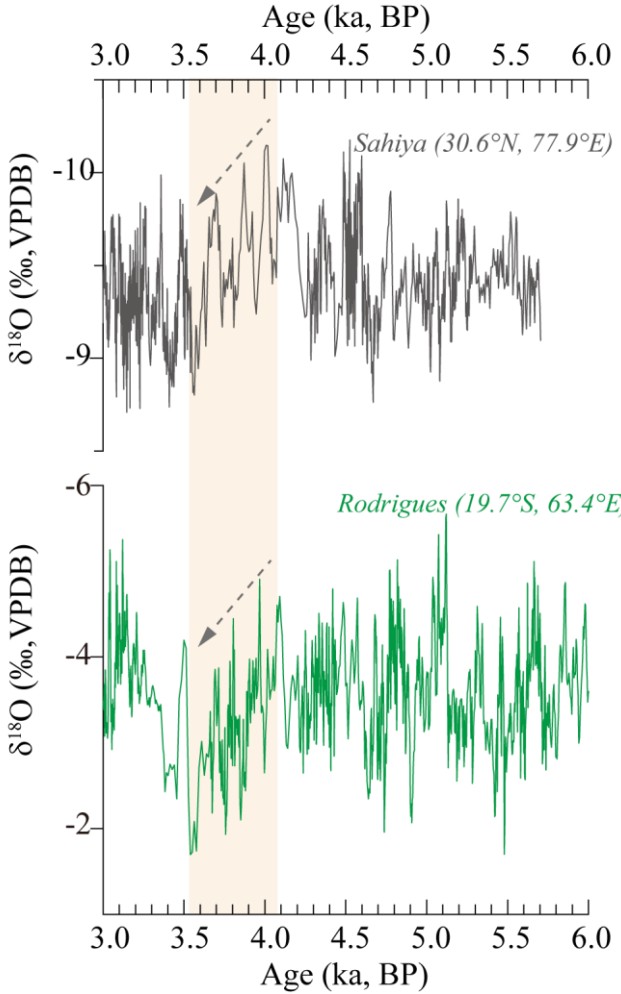

**Figure 9. Comparison of LAVI-4 with the Sahiya cave record.** From top to bottom, speleothem $\delta^{18}O$ record from Sahiya cave, North India (Kathayat et al., 2017). Shaded bar marks the duration from ~ 4 to 3.5 ka BP. Grey dashed arrow line mark the drying trend inferred from both records.