# Peer review of "Hydroclimatic variability in the southwestern Indian Ocean between 6000 and 3000 years ago"

_Climate of the Past, 2018_

## Referee Comment (RC1) · N. Scroxton (Referee) · 10 Sep 2018

General Comments In this high-quality paper Li et al. produce a much-needed record of monsoonal variability during the middle-late Holocene from the southern hemisphere. It fills an important spatial gap in our understanding of the 4.2 kyr BP event, and an important temporal gap of the middle Holocene from the southern Indian Ocean. Given the recent announcement of the Meghalayan age, this paper is also particularly timely. The fundamental conclusion of an expanded and contracted ITCZ during the middle Holocene rather than north-south translation is well founded and an important advance. It may not provide brand new concepts or methods, but not all papers have to do so to be important, and this paper still makes a substantial contribution to progress given the spatial and temporal gaps it fills. It is certainly a suitable paper for Climate of the

[Figure]

Past. I would rate it as excellent under scientific significance.

The record itself is of high-quality, with good replication from a nearby cave and a precise age model. The interpretation of the proxy is well anchored in modelled and observational data. There are a few inconsistencies within the paper (dealt with in the specific comments), but on the whole discussion of the results is balanced, without overreach. The conclusion reached, that of an expansion and contraction of ITCZ during the middle Holocene rather than north-south translation is well founded in places, but perhaps more could be done demonstrate whether this mechanism is functional throughout the entire 3000 year stalagmite growth phase rather than just a 500 year window.

However, I do not believe that one of the conclusions reached is supported by the data and believe there is a temporal mismatch between the interpretation, and what is going on. The authors correctly identify a drying trend that begins at 4.0 or 3.9 kyr and goes on to 3.5 kyr BP. They identify a regional coherency with other speleothem records at Liang Luar, Sahiya and possibly Tangga, and they suggest a plausible mechanism of expansion and contraction of the ITCZ range. However, they erroneously attribute this to the 4.2 kyr event (4.2 kyr BP-3.9 kyr BP) ignoring a small negative isotope excursion (wetter conditions) that exists in the new record and in the Liang Luar and Sahiya records. The regional replication of the signal and quality of the dating on this record and others is sufficient that this offset is unlikely to be due to age model errors. At this time, I would only rate the scientific quality as fair, but I do not believe much work is necessary for it to be excellent.

The paper is well presented and compact, focusing only on what's necessary, with few needless additions. It is well-written in understandable English. Altogether the writing is compact, although there may be one or two too many figures. Of recent papers I have reviewed, this one was the most enjoyable to read. I would rate it as excellent under presentation quality.

Specific Comments Where does the drying start? 4.2 kyr BP as the onset of the drying trend: Line 212 you say that the 4.2kyr event marks the onset of the drying trend. You also use other records as evidence for this around line 236 4.1 kyr BP as the onset of the drying trend: figure 9 deriving from an early d18O low 4.0 kyr BP as the onset of the drying trend: Lines 28, 204, 233, deriving from a late d18O low. Figure 8 3.9 kyr BP as the onset of the drying trend: Lines 27, 214, deriving from the change in mean state

When using 4.1 or 4.0 you measure from the lowest low d18O value (149.2mm or 145.2mm) of one of your two stalagmites. But that doesn't necessarily mean that's the point at which the climate changes. To say drying begins at the wettest part of a wet excursion is technically correct, but misleading. That's just the wettest point of a wet period. I'd be more convinced by the point at which the mean state shifts. Tools such as Rampfit or Bayesian Change Point Analysis would identify this quantitatively, but it qualitatively looks like 3.9 kyr BP to me. In the data there is abrupt shift from d18O values between 3and 4 per mill to d18O values between 2 and 3 per mill at 140.6mm (3887 kyr BP COPRA or 3934 kyr BP ISCAM) with the d13C changing 1mm earlier at 141.6mm (3904/3964 kyr BP). This is also the point at which the mean d18O switches to dry conditions (positive z-score) in figure 6. To me. 3.9 is a more convincing point to start taking about dry periods and drought (especially as the title is about Megadrought).

Is this the 4.2 kyr event or not? Given the age uncertainties on both the event as recorded elsewhere (maybe not great in individual records but I think the community is now pretty satisfied with 4.2 given the array of evidence) and in the stalagmite itself (excellent as expected) then 3.9-3.5 spell of dry conditions is not the 4.2-3.9kyr event. Even if you disagree with my argument above and take 4.0 as the onset of drying, I wouldn't necessarily call that coherence with 4.2 either. As you state in the abstract, the inferred hydroclimatic state over the length of the record is not distinguishable from the region's mean hydroclimate between 4.2 and 4.0 kyr BP.

When you then compare the new record with others, there's a regional coherency to this weak wet and strong dry phasing. Rodrigues has a small wet excursion (likely insignificant as stated in the abstract) between 4.2 and 4.0 kyr BP. You show this explicitly in figure 8. Figure 8 also shows the Liang Luar Record with a wet excursion during this period and drying after. Tangga Cave also possibly shows this pattern too (though its more ambiguous and maybe not quite robust enough to call 'wet'). In Figure 9 you also have Sahiya cave showing a wet excursion between 4.2 and 4.0/3.9 kyr BP, and then drying after*. Three or four precisely dated speleothem records, east and west of the Indian Ocean, north and south of the equator. All showing the same thing. Slightly, but not abnormally, wet conditions (4.2-4.0 kyr BP), dry conditions (4.0-3.5 kyr BP). In which case, does this paper show how unimportant the 4.2 kyr event is in this part of the world. I think you've got a great story here, and it's not being told.

* Kathyat et al., 2017 make the same interpretation here too, analaysing the onset of peak wet conditions as the start of a drying trend, instead of interpreting a wet period followed by a dry period.

What about other parts of the record? Section 4.3 is title about patterns of hydroclimate variability between 6 and 3 kyr BP. Yet the discussion focuses only on a seven-hundred-year period (4.2-3.5 kyr BP). While the proposed mechanism is plausible for this period, is it also plausible for the other 2300 years of speleothem record? I feel this deserves more discussion.

Additional Comments Line 134: Any justification for choosing the standard crustal value for the initial 230/232 ratio? Stratigraphic constraints, isochrons etc.

Line 144: Please state which age model do you end up using as your data?

Line 208: Could gradual positive trends and abrupt terminations be related to drip dynamics rather than climate? A gradually drying karst storage component with rapid fill would produce this kind of shape.

Line 217: On line 217 you state that the Lake Malawi record shows a weak dry excursion 4.1 – 3.5 kyr BP. On Line 254 you state that the Lake Malawi record shows virtually unchanged hydroclimate conditions. Figure 7 looks like very little change in the Lake Malawi record to me. You should probably delete the Line 217 sentence.

Line 258: I understand the desire to include brand new records in a paper, and that these sometimes have to be added last minute. But you introduce an entirely new concept (Southern Hemisphere Westerly winds and their control of restricting the southern range of the ITCZ) in the last sentence of the paper. You either need to introduce this concept much earlier or remove this record and discussion.

Various: Please be consistent with how you label and refer to different records between figures and text. It makes it much harder to follow your argument when you cannot easily switch between the two. Non-speleothem experts who do not know which country relates to which specific cave name and which specific stalagmite in that cave will find it difficult to keep track Line 215: Tatos Basin in text, Mauritius in figure 7 Line 220: Sumatra in text, Tangga cave in figure 8 Line 220: Northwest Australia in text, KNI-51 in figure 8

Figures: 9 figures is a lot for a short paper such as this. Could figures 8 and 9 be combined given they essentially show the same thing – i.e. regional coherency.

Technical Corrections Line 119: No need to use respectively twice in consecutive sentences Line 142: Spell out ISCAM at first usage. Line 172: No need to spell out ISCAM at second usage. Line 166: The Dorale and Liu test is even more convincing when consistent between two nearby caves, you should state this more explicitly that "speleothems from the same cave" Line 173: In line 70, you state PATA1 stops growing at 3.5 kyr BP. In line 173 you have overlap between the two stalagmites up to 3.4 kyr BP. Line 317: Th in superscript Line 349: Blank line? Figure 1: Coastlines should be in a darker color to make them clearer Figure 1: You should show the isohyet on the scale bar Line 497: 199?

---

## Referee Comment (RC2) · Anonymous Referee #2 · 17 Sep 2018

*Climate of the Past* Submission **cp-2018-100,** "Speleothem Evidence for Megadroughts in the SW Indian Ocean during the Late Holocene" by Li et al., presents evidence from two stalagmites from Rodrigues Island in the southwestern Indian Ocean and makes inferences about climate from 4500 to 3000 years BP. The most profound inference is of a "multicentennial period of drought (i.e., megadrought) that lasted continuously from ~ 3.9 to 3.5 ka BP".

        The fundamentals of this kind of research are
               (A) the stalagmite(s) studied,
               (B) the radiometric ages,
               (C) the age model resulting from A and B,
               (D) the data placed in time-series using C, and
               (E) the reasoning employed to interpret the data in D.
I will therefore proceed through this list and then consider the broader implications of the manuscript. This leads to six enumerated suggestions for improvement of the manuscript. A figure on the second page illustrates some of the points made.

**A. The stalagmites studied**
        The study draws on two stalagmites, LAVI-4 and PATA-1, from two caves on Rodrigues Island. I *infer* from Figure 4B that the project does not draw on the entirety of Stalagmite PATA-1. As I said, I had to *infer* this, and I do not see it stated in the text. I think it should be, to save readers confusion, and hence my first suggestion:

**Suggestion 1.** The manuscript and its figures should make explicit what portions of the two stalagmites were analyzed for this project.

        The images of the stalagmites provided in Figure 4 suggest that significant layer-bounding surfaces (Railsback et al., 2013) may be present, but no mention of layer-bounding surfaces is made in the text, leaving the reader to wonder if the authors infer none or, alternately, did not look for them. This leads me to Suggestion 2.

**Suggestion 2:** The manuscript should report the layer-bounding surfaces seen in the stalagmite or state explicitly that there are none.

        Both of these suggestions will matter considerably later.

**B. Radiometric ages**
        The manuscript focuses on the period from 4500 to 3000 years BP (ostensibly around 4.0 ka for the "4.2 ka BP event", but the big result comes later in the 3000s years BP). The number of ages reported is as follows:

| Stalagmite | Number of ages between 4500 and 3000 BP |
|---|---|
| LAVI-4 | 16 |
| PATA-1 | **1** |

        LAVI-4 is clearly thoroughly dated, but PATA-1 has only one age in the time interval of interest. This will be important in Part C and will lead to Suggestion 3.

[Figure]

**(A)** *Possible layer-bounding surface?*

0 20 40 60 80 100 120 140 150 160 180 200 220 240 260 2?0

2cm

**(B)** *Analyzed interval of PATA-1*

0 20 40 60 80 100 120 140

2cm

**LAVI-4**

*Suggested hiatus*

— COPRA
— ISCAM

Age (ka, BP)

**PATA-1**

*Fast rate*

*Slow rate*

— COPRA
— ISCAM

Distance from top (mm)

**(A)** Age (ka, BP)

3.0  3.5  4.0  4.5  5.0  5.5  6.0

*Suggested hiatus*

*3.93 ka BP*  *Two wet pulses*  *4.15 ka BP*

LAVI-4 δ¹⁸O (‰)

PATA-1 δ¹⁸O (‰)

*Why does the PATA-4 record stop here?*

**3.93 ka BP** : **Two wet pulses** : **4.15 ka BP**

**(B)** Age (ka, BP)

3.0  3.5  4.0  4.5  5.0  5.5  6.0

*Only age in PATA-1 across 1700 years*

LAVI-4 δ¹³C (‰)

PATA-1 δ¹³C (‰)

*Improbably instantaneous shift in δ¹³C*

*Why does the PATA-1 record stop here?*

**C. Age models**

**C1: The PATA-1 age model**
Through the time interval of interest, the age model for PATA-1 in Figure 4B is a straight line, and the very quantitative algorithms used to generate the age models give relatively small uncertainties. However, the age of material from 18 to 24 mm from the top is unconstrained because there are no radiometric ages in that interval. Application of growth rates derived from earlier parts of the stalagmite (my dashed lines on Figure 4B) suggests that the material at 22 mm from the top could be anywhere from 4600 to 3900 years old (and a hiatus could make the range even greater).

To summarize Section B and the previous paragraph, because the PATA-1 record from 4.6 to 3.6 ka has only one U-series date, age is largely unconstrained in that interval. Thus PATA-1 provides an isotopic record correlative with that of LAVI-4, but PATA-1 is of no help with chronology. This leads me to Suggestion 3.

**Suggestion 3.** The statements that the manuscript presents "chronologically well-constrained speleothem oxygen and carbon isotopes record**s** of hydroclimate" (Lines 23 and 24) and "**two** precisely dated speleothem oxygen ($\delta^{18}O$) and carbon ($\delta^{13}C$) isotope record**s**" (Line 67) should be changed to the singular "record", and the word "two" should be deleted, because the manuscript in fact presents only one chronologically well-constrained record of the interval of interest, not two. The plural claim "record**s** have tight age control" is likewise invalidated.

**C2: The LAVI-4 age model**
The LAVI-4 age model in Figure 4A presents the results of many radiometric analyses, which is good. Figure 4A suggests a relatively constant rate of growth, with one exception about which three points can be made:
a) Growth is relatively constant except at 123 mm, where the growth rate is much less, suggestive of a hiatus.
b) The image of stalagmite nested in Figure 4A (the only image provided) is not particularly clear, and the indexing scheme is not shown, but my attempt to reconstruct the indexing to which readers are not privy suggests that there may be a layer-bounding surface (and thus a possible hiatus) at about 123 mm.
c) In the stable isotope data from about 123 mm, there is a shift in $\delta^{13}C$ of about 6.2‰ between two successive stable isotope samples. The manuscript says that resolution of the data is about 4 years, so that the data imply a change in $\delta^{13}C$ of about 6.2‰ over about 4 years. That implies a major ecosystem shift and shift in soil carbon (which has decades-to-centuries residence times) in just four years. A more likely explanation is a hiatus in which the unrecorded time allowed the shift in soil ecology at feasible rates.

Points a, b, and c lead to Suggestion 4.

**Suggestion 4.** Either the manuscript should be revised to use an age model including the hiatus evident at about 123 mm, or the manuscript should explicitly explain why it rejects the hiatus that will be evident to many readers. Clearly the statement in Lines 120 to 121 that "both samples grew continuously between 3.5 and 6.0 ka BP interval without any visible hiatuses" should be reconsidered.

As an aside, I would add that the problem here is a common result of generation of age models using computer programs that are not written to include hiatuses and that do not consider evidence beyond the radiometric dates. Use of these programs seems very quantitative and objective, so it is attractive, but it also leads to non-recognition of hiatuses and thus flawed age models. It is far better to recognize a hiatus, to generate a better age model, and to interpret the

cause of the hiatus – and in this case the hiatus is very convincing evidence of extremely dry conditions.

**D. Stable isotope data**

The manuscript reports both $\delta^{18}O$ and $\delta^{13}C$ data, which is good – some researchers oddly do not report their $\delta^{13}C$ data, despite its usefulness. The data reported seem quite normal: range of $\delta^{13}C$ is greater than that of $\delta^{18}O$, both are in the negative single-digit values (relative to VPDB) typical of stalagmites, etc. The two co-vary, which is typical of settings in which rainfall limits the extent of vegetation. LAVI-4 has greater ranges of both $\delta^{13}C$ and $\delta^{18}O$ than PATA-1.

One notable omission is that the stable isotope data from PATA-1 stop at 15 mm below the top of the stalagmite, during the most extreme part of the "megadrought", and do not record the return to the less extreme conditions. This is like reading a novel only to find the last few pages have been torn out. Definition of the time and duration of the megadrought would seemingly require continuation of the PATA-1 series of stable isotope data later above 15 mm from the top of the stalagmite.

Note that the unexplained absence of data from above 15 mm in PATA-1 invalidates the abstract's claim to "present high-resolution and chronologically well-constrained speleothem oxygen and carbon isotopes records of hydroclimate variability **between ~6 and 3 ka ago** from Rodrigues Island": PATA-1 was not analyzed to give a record after 3.5 ka.

Suggestion 5: The PATA-1 series of stable isotope data should be extended higher/later than its present extent, or the manuscript should explain the omission to readers who wonder why it was terminated in mid-event. If the omission persists, the abstract's claim to "record**s** [plural] of hydroclimate variability between ~6 and 3 ka ago" should be modified, because only one record goes to 3 ka.

**E. Reasoning employed to interpret the data**

Lines 105 to 109 lay out the mindset used to interpret the oxygen isotope data, which hinges entirely on the amount effect giving an inverse relationship between $\delta^{18}O$ and rainfall. I know little about the Indian Ocean but find no problem with that general assumption, but there is a growing literature suggesting that post-rainfall effects like evaporation are important too. No literature is cited in the manuscript. McDermott (2004) and Lachniet (2009) commonly are cited with regard to the amount effect and Cuthbert et al. (2014), Markowska et al. (2016), and Treble et al. (2017) are examples of the newer literature.

The rationalization of the carbon isotope data appears much later, in Lines 190 to 195, and similarly seems sound.

**Broader considerations**

The 4.2 ka BP event is prominent in the abstract and introduction, but it hardly gets a mention thereafter.  The former, rather than the latter, seems strange, because the present manuscript is mostly concerned with a major dry event that happened later, at 3.9 to 3.5 ka.  Lines 209 to 211 in fact disavow any recognition of the 4.2 ka BP event, stating the "the interval corresponding to the '4.2 ka event', typically considered between 4.2 and 3.9 ka BP (e.g., Weiss et al., 2016), in the LAVI-4 records does not however, stand out as 'pulse-like' event as evident in many other proxy records".  One thus has to wonder why all the mention of the 4.2 ka BP event in the abstract and introduction.

With that said, one can return to the LAVI-4 data in which age is well constrained and see two negative/wet spikes in $\delta^{18}O$ in the interval from 4.15 to 3.93 ka (see my mark-up figure).  Railsback et al. (2018) concluded that the so-called 4.2 ka BP event took place from 4.15 to 3.93 ka, commonly is recognized as two pulses, and in Namibia can be recognized as two moderately wet pulses.  That's exactly what can be seen in LAVI-4.

This leads to Suggestion 6:

Suggestion 6: Either the manuscript's presently incongruous early focus on the so-called 4.2 ka BP event should be de-emphasized, most notably in the abstract, or the manuscript should discuss the project's data about the 4.2 ka BP event, which suggest a pair of wet pulses congruent with other published data from the Southern Hemisphere's zone of summer rainfall.

**Minor things**

a)  In Line 177, PATA1 should be PATA-1      b) de Boer et al. (2013, 2014, 2015) are listed in the references as "Boer", E.J.D., 2013 . . .", which left this reader scrambling.
c)  Edwards et al. (1987) is between the Cs and Ds in the references.

**References**

Cuthbert, M.O., Baker, A., Jex, C.N., Graham, P.W., Treble, P.C., Andersen, M.S., Acworth, R.I., 2014. Drip water isotopes in semi-arid karst: implications for speleothem paleoclimatology. Earth Planet. Sci. Lett. 395, 194–204.

Lachniet, M.S., 2009. Climate and environmental controls on speleothem oxygen-isotope values. Quat. Sci. Rev. 28, 412–432.

Markowska, M., Baker, A., Andersen, M.S., Jex, C.N., Cuthbert, M.O., Rau, G.C., Graham, P.W., Rutlidge, H., Mariethoz, G., Marjo, C.E., Treble, P.C., Edwards, N., 2016. Semiarid zone caves: evaporation and hydrological controls on $\delta^{18}O$ drip water composition and implications for speleothem paleoclimate reconstructions. Quat. Sci. Rev. 131, 285–301.

McDermott, F., 2004. Palaeo-climate reconstruction from stable isotope variations in speleothems: a review. Quat. Sci. Rev. 23, 901–918.

Railsback, L.B., Akers, P.D., Wang, L., Holdridge, G.A., Voarintsoa, N., 2013. Layer-bounding surfaces in stalagmites as keys to better paleoclimatological histories and chronologies. Int. J. Speleol. 42, 167–180.

Railsback, L.B., Liang, F., Brook, G.A., Voarintsoa, N.R.G., Sletten, H.R., Marais, E., Hardt, B., Cheng, H., Edwards, R.L., 2018. The timing, two-pulsed nature, and variable climatic expression of the 4.2 ka event: A review and new high-resolution stalagmite data from Namibia, Quaternary Science Reviews, 186, 78-90.

Treble, P.C., Baker, A., Ayliffe, L.K., Cohen, T.J., Hellstrom, J.C., Gagan, M.K., Frisia, S., Drysdale, R.N., Griffiths, A.D., Borsato, A., 2017. Hydroclimate of the Last Glacial Maximum and deglaciation in southern Australia's arid margin interpreted from speleothem records (23–15 ka). Clim. Past 13, 667–687.

---

## Author Response (AR1)

**Twin Cities Campus**

**Department of Earth Sciences**
Newton Horace Winchell School of Earth Science

150 John T. Tate Hall
116 Church Street SE
Minneapolis, MN 55455-0219

612-624-1333
Fax: 612-625-3819
Geology@tc.umn.edu
http://www.geo.umn.edu

Nov 04, 2018

Dr. Raymond Bradley
Senior Editor
Climate of the Past
Climate System Research Center
University of Massachusetts
Amherst, MA 01003, USA
rbradley@geo.umass.edu

Dear Raymond,

Thanks so much for providing editorial direction and soliciting excellent comments from two reviewers. We think this process has resulted in an improved manuscript. We used your overview as a guide and have followed through on the overwhelming majority of suggestions/comments. You can follow our point-by-point responses to reviewers' suggestions/comments, a list of changes and a marked-up manuscript (attached at the back). In the revised manuscript, we changed the title of the manuscript into 'Hydroclimatic variability in the southwestern Indian Ocean between 6000 and 3000 years ago'. Due to mismatch between "Late Holocene" in the previous title and the discussed time interval (between 6 and 3 ka BP) in the manuscript, we would like to rename it clearly to avoid confusion.

We very much appreciate your efforts, as well as those of the referees. We hope that we have satisfied the essence of the comments and suggestions.

Sincerely,

Hai Cheng
Professor
Institute of Global Environmental Change
Xi'an Jiaotong University
Xi'an 710049, China
cheng021@mail.xjtu.edu.cn

Senior Research Scientist
Department of Earth Sciences
University of Minnesota
Minneapolis, MN 55455, USA
cheng021@umn.edu

*1.Point-by-point response*
*Reviewer #1* from Dr. Nick Scroxton:
**General Comments:**
In this high-quality paper Li et al. produce a much-needed record of monsoonal variability during the middle-late Holocene from the southern hemisphere. It fills an important spatial gap in our understanding of the 4.2 kyr BP event, and an important temporal gap of the middle Holocene from the southern Indian Ocean. Given the recent announcement of the Meghalayan age, this paper is also particularly timely. The fundamental conclusion of an expanded and contracted ITCZ during the middle Holocene rather than north-south translation is well founded and an important advance. It may not provide brand new concepts or methods, but not all papers have to do so to be important, and this paper still makes a substantial contribution to progress given the spatial and temporal gaps it fills.
The paper is well presented and compact, focusing only on what's necessary, with few needless additions. It is well-written in understandable English. Altogether the writing is compact, although there may be one or two too many figures. Of recent papers I have reviewed, this one was the most enjoyable to read. I would rate it as excellent under presentation quality.

Answer- We thank the reviewer for his positive evaluation of our manuscript.

It is certainly a suitable paper for Climate of the Past. I would rate it as excellent under scientific significance. The record itself is of high-quality, with good replication from a nearby cave and a precise age model. The interpretation of the proxy is well anchored in modelled and observational data. There are a few inconsistencies within the paper (dealt with in the specific comments), but on the whole discussion of the results is balanced, without overreach. The conclusion reached, that of an expansion and contraction of ITCZ during the middle Holocene rather than north-south translation is well founded in places, but perhaps more could be done demonstrate whether this mechanism is functional throughout the entire 3000 year stalagmite growth phase rather than just a 500 year window.

Answer- Briefly, we find that the ITCZ contraction mechanism best explains the observed temporal patterns of changes in climate inferred from various proxy records only between 4.1 and 3.5 ka. We will add new text in the discussion section throughout the length of our record. Centered with the climatic variations during the 4.2 ka event (between 4.2 and 3.9 ka BP), we will discuss the main pattern of climate variation before and after 4.2 ka event.

However, I do not believe that one of the conclusions reached is supported by the data and believe there is a temporal mismatch between the interpretation, and what is going on. The authors correctly identify a drying trend that begins at 4.0 or 3.9 kyr and goes on to 3.5 kyr BP. They identify a regional coherency with other speleothem records at Liang Luar, Sahiya and possibly Tangga, and they suggest a plausible mechanism of expansion and contraction of the ITCZ range. However, they erroneously attribute this to the 4.2 kyr event (4.2 kyr BP-3.9 kyr BP) ignoring a small negative isotope excursion (wetter conditions) that exists in the new record and in the Liang Luar and Sahiya records. The regional replication of the signal and quality of the dating on this record and others is sufficient that

this offset is unlikely to be due to age model errors. At this time, I would only rate the scientific quality as fair, but I do not believe much work is necessary for it to be excellent.

Answer- This is a good comment/suggestion. We did not mean to attribute the 3.9 to 3.5 ka drought to the 4.2 ka event but perhaps we did not make it clear. We will clarify the discussion in the revised text and figures.

**Specific Comments**:
Where does the drying start? 4.2 kyr BP as the onset of the drying trend: Line 212 you say that the 4.2kyr event marks the onset of the drying trend. You also use other records as evidence for this around line 236. 4.1 kyr BP as the onset of the drying trend: figure 9 deriving from an early d18O low 4.0 kyr BP as the onset of the drying trend: Lines 28, 204, 233, deriving from a late d18O low. Figure 8 3.9 kyr BP as the onset of the drying trend: Lines 27, 214, deriving from the change in mean state.
When using 4.1 or 4.0 you measure from the lowest low d18O value (149.2mm or 145.2mm) of one of your two stalagmites. But that doesn't necessarily mean that's the point at which the climate changes. To say drying begins at the wettest part of a wet excursion is technically correct, but misleading. That's just the wettest point of a wet period. I'd be more convinced by the point at which the mean state shifts. Tools such as Rampfit or Bayesian Change Point Analysis would identify this quantitatively, but it qualitatively looks like 3.9 kyr BP to me. In the data there is abrupt shift from d18O values between 3and 4 per mill to d18O values between 2 and 3 per mill at 140.6mm (3887 kyr BP COPRA or 3934 kyr BP ISCAM) with the d13C changing 1mm earlier at 141.6mm (3904/3964 kyr BP). This is also the point at which the mean d18O switches to dry conditions (positive z-score) in figure 6. To me. 3.9 is a more convincing point to start taking about dry periods and drought (especially as the title is about Megadrought).

Answer- We agree with the reviewer that the onset of the drying trend should be considered at a point where both $\delta^{18}O$ and $\delta^{13}C$ values increased above the long-term mean of the record. We confirmed that transition occurred ~3.9 ka BP by using change-point function in RAMPFIT (Mudelsee, 2000).

Is this the 4.2 kyr event or not? Given the age uncertainties on both the event as recorded elsewhere (maybe not great in individual records but I think the community is now pretty satisfied with 4.2 given the array of evidence) and in the stalagmite itself (excellent as expected) then 3.9-3.5 spell of dry conditions is not the 4.2-3.9kyr event. Even if you disagree with my argument above and take 4.0 as the onset of drying, I wouldn't necessarily call that coherence with 4.2 either. As you state in the abstract, the inferred hydroclimatic state over the length of the record is not distinguishable from the region's mean hydroclimate between 4.2 and 4.0 kyr BP.

Answer- The reviewer is right. As we noted in the previous response, we will refine in the revised version that the dry event between 3.9 and 3.5 ka BP is temporally a 'post-event' megadrought of the 4.2 ka event.

When you then compare the new record with others, there's a regional coherency to this weak wet and strong dry phasing. Rodrigues has a small wet excursion (likely insignificant as stated in the abstract) between 4.2 and 4.0 kyr BP. You show this explicitly in figure 8. Figure 8 also shows the Liang Luar Record with a wet excursion during this period and drying after. Tangga Cave also possibly shows this pattern too (though its more ambiguous and maybe not quite robust enough to call 'wet'). In Figure 9 you also have Sahiya cave showing a wet excursion between 4.2 and 4.0/3.9 kyr BP, and then drying after*. Three or four precisely dated speleothem records, east and west of the Indian Ocean, north and south of the equator. All showing the same thing. Slightly, but not abnormally, wet conditions (4.2-4.0 kyr BP), dry conditions (4.0-3.5 kyr BP). In which case, does this paper show how unimportant the 4.2 kyr event is in this part of the world. I think you've got a great story here, and it's not being told.

* Kathyat et al., 2017 make the same interpretation here too, analaysing the onset of peak wet conditions as the start of a drying trend, instead of interpreting a wet period followed by a dry period.

What about other parts of the record? Section 4.3 is title about patterns of hydroclimate variability between 6 and 3 kyr BP. Yet the discussion focuses only on a seven-hundredyear period (4.2-3.5 kyr BP). While the proposed mechanism is plausible for this period, is it also plausible for the other 2300 years of speleothem record? I feel this deserves more discussion.

Answer- Following the comments/suggestions, we will substantially revise our text to address most of the reviewer's comments above: we characterize the main pattern of climate variability in terms of three parts, including the pre-event, '4.2ka event' and the post-event.

Additional Comments Line 134: Any justification for choosing the standard crustal value for the initial 230/232 ratio? Stratigaphic constraints, isochrons etc.

Answer- The initial $^{230}$Th correction of all dates assumes an initial $^{230}$Th/$^{232}$Th atomic ratio of $(4.4 \pm 2.2) \times 10^{-6}$ (the crustal value). We arbitrarily assigned a large uncertainty to the initial ratio, i.e., 50%. The fact that all the dates with different measured $^{230}$Th/$^{232}$Th ratios are in stratigraphic order after corrections is consistent with the assumption for the correction. In addition, most corrections result in relatively small changes due to low $^{232}$Th contents.

Line 144: Please state which age model do you end up using as your data?

Answer- We used the COPRA program to obtain our age model. We will add this information in the revised manuscript.

Line 208: Could gradual positive trends and abrupt terminations be related to drip dynamics rather than climate? A gradually drying karst storage component with rapid fill would produce this kind of shape.

Answer- We will address this comment by adding a new paragraph to discuss the possibility of the effect from the soil zone and epikarst, in addition to rainfall amount.

Line 217: On line 217 you state that the Lake Malawi record shows a weak dry excursion 4.1 – 3.5 kyr BP. On Line 254 you state that the Lake Malawi record shows virtually unchanged hydroclimate conditions. Figure 7 looks like very little change in the Lake Malawi record to me. You should probably delete the Line 217 sentence.

Answer- We agree and delete.

Line 258: I understand the desire to include brand new records in a paper, and that these sometimes have to be added last minute. But you introduce an entirely new concept (Southern Hemisphere Westerly winds and their control of restricting the southern range of the ITCZ) in the last sentence of the paper. You either need to introduce this concept much earlier or remove this record and discussion.

Answer- We agree. We will remove the discussion about the Southern Hemisphere Westerlies in the revised version.

Various: Please be consistent with how you label and refer to different records between figures and text. It makes it much harder to follow your argument when you cannot easily switch between the two. Non-speleothem experts who do not know which country relates to which specific cave name and which specific stalagmite in that cave will find it difficult to keep track Line 215: Tatos Basin in text, Mauritius in figure 7 Line 220: Sumatra in text, Tangga cave in figure 8 Line 220: Northwest Australia in text, KNI-51 in figure 8.

Answer- We will uniform the names.

Figures: 9 figures is a lot for a short paper such as this. Could figures 8 and 9 be combined given they essentially show the same thing – i.e. regional coherency.

Answer- According to reviewer's suggestion, we will move the previous Figures 2 to 5 to supplementary materials to reduce the number of figures in the main text. We will also combine the previous Figures 7 and 9 into one figure, i.e., Figure 5 in the revised manuscript.

Technical Corrections Line 119: No need to use respectively twice in consecutive sentences. Line 142: Spell out ISCAM at first usage. Line 172: No need to spell out ISCAM at second usage. Line 166: The Dorale and Liu test is even more convincing when consistent between two nearby caves, you should state this more explicitly that "speleothems from the same cave" Line 173: In line 70, you state PATA1 stops growing at 3.5 kyr BP. In line 173 you have overlap between the two stalagmites up to 3.4 kyr BP. Line 317: Th in superscript Line 349: Blank line? Figure 1: Coastlines should be in a darker color to make them clearer Figure 1: You should show the isohyet on the scale bar Line 497: 199?

Answer- We agree and will correct in the revised version.

*Reviewer #2:*

*Climate of the Past* Submission **cp-2018-100,** "Speleothem Evidence for Megadroughts in the SW Indian Ocean during the Late Holocene" by Li et al., presents evidence from two stalagmites from Rodrigues Island in the southwestern Indian Ocean and makes inferences about climate from 4500 to 3000 years BP. The most profound inference is of a "multicentennial period of drought (i.e., megadrought) that lasted continuously from ~ 3.9 to 3.5 ka BP".

The fundamentals of this kind of research are

(A) the stalagmite(s) studied,

(B) the radiometric ages,

(C) the age model resulting from A and B,

(D) the data placed in time-series using C, and

(E) the reasoning employed to interpret the data in D.

I will therefore proceed through this list and then consider the broader implications of the manuscript. This leads to six enumerated suggestions for improvement of the manuscript. A figure on the second page illustrates some of the points made.

**A. The stalagmites studied**

The study draws on two stalagmites, LAVI-4 and PATA-1, from two caves on Rodrigues Island. I *infer* from Figure 4B that the project does not draw on the entirety of Stalagmite PATA-1. As I said, I had to *infer* this, and I do not see it stated in the text. I think it should be, to save readers confusion, and hence my first suggestion:

**Suggestion 1.** The manuscript and its figures should make explicit what portions of the two stalagmites were analyzed for this project.

Answer- We agree. We will add explicit descriptions about the portions of the two stalagmite that we used in this study. We will clearly mark the portions of the two samples using blue bars in the related figure.

The images of the stalagmites provided in Figure 4 suggest that significant layerbounding surfaces (Railsback et al., 2013) may be present, but no mention of layer-bounding surfaces is made in the text, leaving the reader to wonder if the authors infer none or, alternately, did not look for them. This leads me to Suggestion 2.

**Suggestion 2:** The manuscript should report the layer-bounding surfaces seen in the stalagmite or state explicitly that there are none.

Both of these suggestions will matter considerably later.

Answer- Thanks for the suggestion. Railsback et al. (2013) identified two types of layer-bounding surfaces in their stalagmite: Type E, formed under wet conditions and Type L reflecting dry conditions. Upon a closer petrographic inspection, type L surface likely occurred at ~124 mm depth in stalagmite LAVI-4.

**B. Radiometric ages**

The manuscript focuses on the period from 4500 to 3000 years BP (ostensibly around 4.0 ka for the "4.2 ka BP event", but the big result comes later in the 3000s years BP). The number of ages reported is as follows:

Stalagmite     Number of ages between 4500 and 3000 BP

|          |    |
|----------|----|
| LAVI-4   | 16 |
| PATA-1   | **1** |

LAVI-4 is clearly thoroughly dated, but PATA-1 has only one age in the time interval of interest. This will be important in Part C and will lead to Suggestion 3.

Answer- In order to improve the age model for sample PATA-1 we have obtained two additional $^{230}$Th dates between 4 and 4.5 ka BP. The new subsamples of PATA-1 were drilled at 20 mm (4284 ±87 years BP) and 22.2 mm (4494 ±138 years BP), respectively. The reconstructed age model with new additional dates is consistent with the previous one, but more robust.

**C. Age models**
**C1: The PATA-1 age model**
Through the time interval of interest, the age model for PATA-1 in Figure 4B is a straight line, and the very quantitative algorithms used to generate the age models give relatively small uncertainties. However, the age of material from 18 to 24 mm from the top is unconstrained because there are no radiometric ages in that interval. Application of growth rates derived from earlier parts of the stalagmite (my dashed lines on Figure 4B) suggests that the material at 22 mm from the top could be anywhere from 4600 to 3900 years old (and a hiatus could make the range even greater).

To summarize Section B and the previous paragraph, because the PATA-1 record from 4.6 to 3.6 ka has only one U-series date, age is largely unconstrained in that interval. Thus PATA-1 provides an isotopic record correlative with that of LAVI-4, but PATA-1 is of no help with chronology. This leads me to Suggestion 3.

**Suggestion 3.** The statements that the manuscript presents "chronologically well-constrained speleothem oxygen and carbon isotopes record**s** of hydroclimate" (Lines 23 and 24) and "**two** precisely dated speleothem oxygen (δ18O) and carbon (δ13C) isotope record**s**" (Line 67) should be changed to the singular "record", and the word "two" should be deleted, because the manuscript in fact presents only one chronologically well-constrained record of the interval of interest, not two. The plural claim "record**s** have tight age control" is likewise invalidated.

Answer- We agree with the reviewer. As we noted in the previous response, we have added two additional dates for PATA-1. We will refine the statements in the revised version.

**C2: The LAVI-4 age model**
The LAVI-4 age model in Figure 4A presents the results of many radiometric analyses, which is good. Figure 4A suggests a relatively constant rate of growth, with one exception about which three points can be made:

a) Growth is relatively constant except at 123 mm, where the growth rate is much less, suggestive of a hiatus.

b) The image of stalagmite nested in Figure 4A (the only image provided) is not particularly clear, and the indexing scheme is not shown, but my attempt to reconstruct the indexing to which readers are not privy suggests that there may be a layer-bounding surface (and thus a possible hiatus) at about 123 mm.

c) In the stable isotope data from about 123 mm, there is a shift in δ13C of about 6.2‰ between two successive stable isotope samples. The manuscript says that resolution of the data is about 4 years, so that the data imply a change in δ13C of about 6.2‰ over about 4 years. That implies a major ecosystem shift and shift in soil carbon (which has decades-to-centuries residence times) in just four years. A more likely explanation is a hiatus in which the unrecorded time allowed the shift in soil ecology at feasible rates. Points a, b, and c lead to Suggestion 4.

**Suggestion 4.** Either the manuscript should be revised to use an age model including the hiatus evident at about 123 mm, or the manuscript should explicitly explain why it rejects the hiatus that will be evident to many readers. Clearly the statement in Lines 120 to 121 that "both samples grew continuously between 3.5 and 6.0 ka BP interval without any visible hiatuses" should be reconsidered.

As an aside, I would add that the problem here is a common result of generation of age models using computer programs that are not written to include hiatuses and that do not consider evidence beyond the radiometric dates. Use of these programs seems very quantitative and objective, so it is attractive, but it also leads to non-recognition of hiatuses and thus flawed age models. It is far better to recognize a hiatus, to generate a better age model, and to interpret the cause of the hiatus – and in this case the hiatus is very convincing evidence of extremely dry conditions.

Answer- Good point again. We will add a figure to show two possibilities for the sample LAVI-4 at a depth of ~124 mm, a hiatus (~100 years) as suggested by the reviewer, or a portion of the sample with a very slow growth rate. Either way, the major hydroclimatic patterns between 6 and 3 ka BP inferred from our records, thus our conclusions, remain similar.

**D. Stable isotope data**
The manuscript reports both δ18O and δ13C data, which is good – some researchers oddly do not report their δ13C data, despite its usefulness. The data reported seem quite normal: range of δ13C is greater than that of δ18O, both are in the negative single-digit values (relative to VPDB) typical of stalagmites, etc. The two co-vary, which is typical of settings in which rainfall limits the extent of vegetation. LAVI-4 has greater ranges of both δ13C and δ18O than PATA-1.

One notable omission is that the stable isotope data from PATA-1 stop at 15 mm below the top of the stalagmite, during the most extreme part of the "megadrought", and do not record the return to the less extreme conditions. This is like reading a novel only to find the last few pages have been torn out. Definition of the time and duration of the megadrought would seemingly require continuation of the PATA-1 series of stable isotope data later above 15 mm from the top of the stalagmite.

Note that the unexplained absence of data from above 15 mm in PATA-1 invalidates the abstract's claim to "present high-resolution and chronologically well-constrained speleothem oxygen and carbon isotopes records of hydroclimate variability **between ~6 and 3 ka ago** from Rodrigues Island": PATA-1 was not analyzed to give a record after 3.5 ka.

**Suggestion 5:** The PATA-1 series of stable isotope data should be extended higher/later than its present extent, or the manuscript should explain the omission to readers who

wonder why it was terminated in mid-event. If the omission persists, the abstract's claim to "record**s** [plural] of hydroclimate variability between ~6 and 3 ka ago" should be modified, because only one record goes to 3 ka.

Answer- Sample PATA-1 shows a major hiatus at 15 mm. Based on our dating results, growth re-commenced after ~630 years at ~2740 yr BP. Thus, the record of the top 15 mm may not be helpful for the issues we addressed here. We will add this information in the revised manuscript.

**E. Reasoning employed to interpret the data**
Lines 105 to 109 lay out the mindset used to interpret the oxygen isotope data, which hinges entirely on the amount effect giving an inverse relationship between δ18O and rainfall. I know little about the Indian Ocean but find no problem with that general assumption, but there is a growing literature suggesting that post-rainfall effects like evaporation are important too. No literature is cited in the manuscript. McDermott (2004) and Lachniet (2009) commonly are cited with regard to the amount effect and Cuthbert et al. (2014), Markowska et al. (2016), and Treble et al. (2017) are examples of the newer literature.
The rationalization of the carbon isotope data appears much later, in Lines 190 to 195, and similarly seems sound.

Answer- According to this suggestion, we will add a paragraph to discuss the effect of prior evaporation process in changing the composition of drip-water $\delta^{18}O$ in the shallow soil zone and epikarst. We will also cite the relevant papers, including those suggested by the reviewer.

**Broader considerations**
The 4.2 ka BP event is prominent in the abstract and introduction, but it hardly gets a mention thereafter. The former, rather than the latter, seems strange, because the present manuscript is mostly concerned with a major dry event that happened later, at 3.9 to 3.5 ka. Lines 209 to 211 in fact disavow any recognition of the 4.2 ka BP event, stating the "the interval corresponding to the '4.2 ka event', typically considered between 4.2 and 3.9 ka BP (e.g., Weiss et al., 2016), in the LAVI-4 records does not however, stand out as 'pulse-like' event as evident in many other proxy records". One thus has to wonder why all the mention of the 4.2 ka BP event in the abstract and introduction.
With that said, one can return to the LAVI-4 data in which age is well constrained and see two negative/wet spikes in δ18O in the interval from 4.15 to 3.93 ka (see my mark-up figure). Railsback et al. (2018) concluded that the so-called 4.2 ka BP event took place from 4.15 to 3.93 ka, commonly is recognized as two pulses, and in Namibia can be recognized as two moderately wet pulses. That's exactly what can be seen in LAVI-4.This leads to Suggestion 6:
**Suggestion 6:** Either the manuscript's presently incongruous early focus on the so-called 4.2 ka BP event should be de-emphasized, most notably in the abstract, or the manuscript should discuss the project's data about the 4.2 ka BP event, which suggest a pair of wet pulses congruent with other published data from the Southern Hemisphere's zone of summer rainfall.

Answer- We addressed this comment by keeping the focus on the 4.2 ka event and using the record before and after the event to provide a climatic context for the event. First, we agree with the reviewer. In the revised manuscript, we will characterize the wet and dry events during the 4.2 ka event (4.2 to 3.9 ka BP), and discuss its correlation with other well-dated records (e.g., the Dante cave record (Railsback et al., 2018)). Then we will point out that the multi-decadal fluctuations during the 4.2 ka event are similar to those in the time period from 6 to 4.2 ka BP with a mean state of our entire record between 6 and 3 ka BP. Third, we'll characterize the aridity between 3.9 and 3.5 ka BP as a 'post-event' megadrought. Thus, our data provide new insights not only into the climatic variability during the 4.2 ka event, but also broader background information surrounding the event.

**Minor things**
a) In Line 177, PATA1 should be PATA-1 b) de Boer et al. (2013, 2014, 2015) are listed in the references as "Boer", E.J.D., 2013 . . .", which left this reader scrambling. c) Edwards et al. (1987) is between the Cs and Ds in the references.
Answer- We agree. We will fix these in the revised manuscript.

**2. A list of changes**
1) We add explicit descriptions about the portions of the two stalagmite that we used in this study. (Lines 153-154 and 159 in Page 4) We clearly mark the portions of the two samples using blue bars in the Supplementary Fig.3.
2) We add the layer-boundary and related age-model discussion for LAVI-4 at 124mm. (Lines 155-158 in Page 4)
3) We add two additional dates for PATA-1 and update its age-model. We clarify a hiatus of PATA-1 occurred around 15 mm. (Lines 159-160 in Page 4)
4) We clarify that we use COPPRA result for further discussion. (Lines 161-162 in Page4)
5) We add the discussion about the effect from the soil zone and epikarst and cite the references mentioned by the reviewer. (Lines 195-202 in Page 5)
6) We clarify the climate variation pattern in terms of pre-event, '4.2 ka event' and post-event between 6000 and 3000yr BP. (Lines 210-213, 217-219 and 225-232 in Page 6)
7) We remove the statement saying that Lake Malawi record shows a weak dry excursion.
8) We remove the discussion about Southern Hemisphere Westerlies.
9) We uniform the names in the text and graphs. (e.g.: Lines 233-238 in Page 6 and Fig.4)
10) We change previous Fig.5 to Fig.2, change previous Fig. 6 to Fig.3, change previous Fig. 8 to Fig. 4 and combine previous Figs. 7 and 9 into Fig. 5. We move the previous Figures 2 to 4 to supplementary materials.

*3. A marked-up manuscript*

[revised manuscript text omitted]
 into the North Atlantic Ocean that could have disrupted the Atlantic meridional overturning circulation (AMOC), and thereby, produced changes in global climate in a manner akin to the 8.2 ka BP event (e.g., Cheng et al., 2009; Walker et al., 2012). There is

also no evidence of major perturbations in atmospheric concentrations of aerosols and $CO_2$ at this time (Monnin et al., 2001). A southward shift of the mean position of the ITCZ during this time has been hypothesized (e.g., Mayewski et al., 2004), which potentially could account for the low-latitude aridity observed in many NH locations, but this hypothesis is inconsistent with proxy records situated at the southern margin of the ITCZ in the SH, which show little or no evidence of presumably wetter conditions resulting from a southward shift of the ITCZ (e.g., Russell et al., 2003; De Boer et al., 2013, 2014, 2015; Rijsdijk et al., 2009, 2011; Berke et al., 2012; Railsback et al., 2018). Other hypotheses call for an onset of the modern El Niño Southern Oscillation (ENSO) regime (e.g., Donders et al., 2008; Conroy et al., 2008; Barron and Anderson, 2010), and/or changes in the sea-surface temperature (SST) gradient between the western and eastern Indian Ocean (the Indian Ocean Dipole, IOD) (Berke et al., 2012; De Boer et al., 2014). If the IOD or ENSO are considered as the main driving mechanisms, an anti-phase relationship between the climate at Rodrigues in the west and that on the eastern margin of the southern Indian Ocean including northern Australia would be expected (see the spatial pattern of El Niño related precipitation anomalies in Supplementary Fig. 1D), which, however, is inconsistent with the phase relationship illustrated in Figure 4. As such, the IOD and ENSO mechanisms do not readily explain the observed climate relationship between Rodrigues and other sites across the southern Indian Ocean.

[revised manuscript text omitted]

**Supplementary Fig. 3. Age models of LAVI-4 and PATA-1 stalagmites. (A and C)** scan pictures of stalagmite LAVI-4 and PATA-1, respectively. The blue bars line on the stalagmite slabs showing the stable isotope tracks. Dash line in **A** marks the layer at 124 mm. Arrow in **C** marks the layer at 15mm. (**B**) LAVI-4 age models and age uncertainties obtained using COPRA (Breitenbach et al., 2012) (red) and ISCAM (Fohlmeister, 2012) (black). The gray band depicts the 95% confidence interval from COPRA. Error bars on [230]Th dates represent 2σ analytical errors. (**D**) Same as in (**B**) but for sample PATA-1.

[Figure]

**Supplementary Fig. 4. Comparison of COPRA age model results. (A and B)** COPRA age models (Breitenbach et al., 2012) of LAVI-4 with a hiatus at 124 mm **(A)** and no hiatus **(B)**. **(C)** $\delta^{18}O$ time series based on the age models in **A** and **B**. **(D)** $\delta^{13}C$ time series based on the age models in **A** and **B**. The blue and red lines are the age model results from **A** and **B**, respectively. There is a small offset between the two models, except for the period between 3.55 and 3.4 ka BP marked by red dashed lines. The main hydroclimate variations between 6 and 3 ka BP are robust irrespective of the age model used.

---

## Editor Decision (ED1)

The revised manuscript appropriately responds to reviewers concerns and suggestions. I have only a few minor requests.

1. In Section 1, paragraph 2, please change, "*The goal of this study is to investigate the '4.2 ka event' in a key region…*"
   to
   "*The goal of this study is to investigate a time period that spans the '4.2 ka event' in a key region…*"
   I make this distinction to make it clear that you have no pre-conceived expectation of a "4.2ka signal" in the record, & are investigating a window of time that includes that interval, to see what occurred in Rodrigues. I think this is important, given that you find nothing particularly unusual at that time in your samples.

2. In Section 5.2 you state, "*In this regard, the 4.2 ka event does not appear to be a strong 'single pulse-like' signal in Rodrigues in the context of the long-term climate variance between 6 and 3 ka BP.…*".
   I think the important point is that there is no obvious signal of a "4.2ka B.P. event" at all, and I think you should state that quite explicitly. It is important that areas where there is no evidence for such an anomaly be identified, so we can constrain the signal and (perhaps) figure out what the possible cause was.
   Accordingly, I suggest that you re-phrase this sentence, as "*In this regard, in the context of the long-term climate variance between 6 and 3 ka BP, there is no evidence for an unusual climatic anomaly between 4.2 and 3.9ka B.P. Consistently…*" etc..

3. Also, it seems odd that, after finding no evidence for a "4.2ka BP event", Section 5.3 begins by discussing the driving mechanisms of this "event". I suggest that you eliminate this paragraph and begin Section 5.3 with the second paragraph, "*A close examination…*" I don't think that the first paragraph adds very much to your paper, given that it addresses something that you did not find!

After these minor changes, I think the paper will be very acceptable for publication in Climate of the Past.

---

## Author Response (AR2)

**UNIVERSITY OF MINNESOTA**

*Twin Cities Campus*

**Department of Earth Sciences**
Newton Horace Winchell School of Earth Science

150 John T. Tate Hall
116 Church Street SE
Minneapolis, MN 55455-0219

612-624-1333
Fax: 612-625-3819
Geology@tc.umn.edu
http://www.geo.umn.edu

Nov 21, 2018

Dr. Raymond Bradley
Senior Editor
Climate of the Past
rbradley@geo.umass.edu

Dear Raymond,

Thanks so much for providing clear editorial direction for our manuscript entitled "*Hydroclimatic variability in the southwestern Indian Ocean between 6000 and 3000 years ago*".  We have revised our manuscript strictly based on your instructions/suggestions.  Here attached are our point-by-point responses, a list of changes and a marked-up manuscript.  In addition, we have checked the manuscript carefully for typos, co-authors' names and their affiliations.  The data in the supplementary tables are the latest version used in the manuscript.

We very much appreciate your tremendous help and effort, as well as those of the referees.  We hope that we have satisfied the essence of the comments and suggestions.

Sincerely,

Hai Cheng
Professor
Institute of Global Environmental Change
Xi'an Jiaotong University
Xi'an 710049, China
cheng021@mail.xjtu.edu.cn

Senior Research Scientist
Department of Earth Sciences
University of Minnesota
Minneapolis, MN 55455, USA
cheng021@umn.edu

**1. Editor requests:**

*1. In Section 1, paragraph 2, please change, "The goal of this study is to investigate the '4.2 ka event' in a key region..." to "The goal of this study is to investigate a time period that spans the '4.2 ka event' in a key region..."*

I make this distinction to make it clear that you have no pre-conceived expectation of a "4.2ka signal" in the record, & are investigating a window of time that includes that interval, to see what occurred in Rodrigues. I think this is important, given that you find nothing particularly unusual at that time in your samples.

*2. In Section 5.2 you state, "In this regard, the 4.2 ka event does not appear to be a strong 'single pulse-like' signal in Rodrigues in the context of the long-term climate variance between 6 and 3 ka BP....".*

I think the important point is that there is no obvious signal of a "4.2ka B.P. event" at all, and I think you should state that quite explicitly. It is important that areas where there is no evidence for such an anomaly be identified, so we can constrain the signal and (perhaps) figure out what the possible cause was. Accordingly, I suggest that you re-phrase this sentence, as *"In this regard, in the context of the long-term climate variance between 6 and 3 ka BP, there is no evidence for an unusual climatic anomaly between 4.2 and 3.9ka B.P. Consistently..." etc..*

3. Also, it seems odd that, after finding no evidence for a "4.2ka BP event", Section 5.3 begins by discussing the driving mechanisms of this "event". I suggest that you eliminate this paragraph and begin Section 5.3 with the second paragraph, *"A close examination..."* I don't think that the first paragraph adds very much to your paper, given that it addresses something that you did not find!

After these minor changes, I think the paper will be very acceptable for publication in Climate of the Past.

**Answer:** Done.

**2. A list of changes**

1) We change the sentence "The goal of this study is to investigate the '4.2 ka event' in a key region…" to "*The goal of this study is to investigate a time period that spans the '4.2 ka event' in a key region...*" (Lines 60-61 in Page 2).

2) We change the sentence "In this regard, the 4.2 ka event does not appear to be a strong 'single pulse-like' signal in Rodrigues in the context of the long-term climate variance between 6 and 3 ka BP…" to "*In this regard, in the context of the long-term climate variance between 6 and 3 ka BP, there is no evidence for an unusual climatic anomaly between 4.2 and 3.9ka BP.*" (Lines 221-225 in Page 6).

3) We remove the first paragraph in section 5.3 and related references.

4) We make some minor corrections for typos, data information and graphs (Figs. 4 and 5: unify the font size in each gragh).

*3. A marked-up manuscript*

[revised manuscript text omitted]

**Supplementary Fig. 3. Age models of LAVI-4 and PATA-1 stalagmites. (A and C)** scan pictures of stalagmite LAVI-4 and PATA-1, respectively. The blue bars line on the stalagmite slabs showing the stable isotope tracks. Dash line in **A** marks the layer at 124 mm. Arrow in **C** marks the layer at 15mm. (**B**) LAVI-4 age models and age uncertainties obtained using COPRA (Breitenbach et al., 2012) (red) and ISCAM (Fohlmeister, 2012) (black). The gray band depicts the 95% confidence interval from COPRA. Error bars on $^{230}$Th dates represent $2\sigma$ analytical errors. (**D**) Same as in (**B**) but for sample PATA-1.

[Figure]

**Supplementary Fig. 4. Comparison of COPRA age model results. (A and B)** COPRA age models (Breitenbach et al., 2012) of LAVI-4 with a hiatus at 124 mm **(A)** and no hiatus **(B)**. **(C)** $\delta^{18}O$ time series based on the age models in **A** and **B**. **(D)** $\delta^{13}C$ time series based on the age models in **A** and **B**. The blue and red lines are the age model results from **A** and **B**, respectively. There is a small offset between the two models, except for the period between 3.55 and 3.4 ka BP marked by red dashed lines. The main hydroclimate variations between 6 and 3 ka BP are robust irrespective of the age model used.